# **Integrating Fire-Induced Meteorological Changes into Plume Rise Modeling for Extreme Wildfire Simulations**

Lisa Janina Muth<sup>1</sup>, Gholam Ali Hoshyaripour<sup>1</sup>, Bernhard Vogel<sup>1</sup>, Heike Vogel<sup>1</sup>, and Corinna Hoose<sup>1</sup> Institute of Meteorology and Climate Research, Karlsruhe Institute of Technology (KIT), Karlsruhe, Germany **Correspondence:** Lisa Janina Muth (lisa.muth@kit.edu)

Abstract. Wildfire emissions are a major environmental concern, especially as climate change increases the frequency of extreme events. Our study investigates the limitations of the widely used Freitas plume-rise model during the Australian New Year's wildfires of 2019/2020, focusing on how accounting for fire-atmosphere feedbacks in the host model affects plume rise. Simulations were conducted at a 6.6 km grid resolution, where convection is parameterized but fire-induced meteorological effects remain significant. Including fire-induced moisture release led to increased cloud formation, but had minimal impact on plume dynamics. In contrast, accounting for fire-induced heat release significantly increased the plume height due to enhanced buoyancy and cloud formation, even without added moisture.

Simulating aerosol-radiation interactions initially reduced injection height, as solar absorption by dense aerosols stabilized the

atmosphere. However, a lofting effect emerged from the second day onward. The combined simulation—incorporating heat and moisture release and aerosol-radiation interaction—produced the highest plume rise and best matched satellite observations, including the aerosol layer in the upper troposphere/lower stratosphere. The effects were strongest on the first day, when fire intensity peaked. For less intense fires, the Freitas plume-rise model performed well without additional feedback mechanisms.

#### 1 Introduction

Wildfires are major sources of aerosols (black carbon (BC) and organic carbon (OC)) and trace gases (mainly CO, CO<sub>2</sub>, and CH<sub>4</sub>), which influence atmospheric composition and climate processes (Galanter et al., 2010). These emissions can be transported over long distances, depending on fire intensity and meteorological conditions, affecting atmospheric chemistry, cloud formation, and radiative forcing on regional to global scales (Val Martin et al., 2006).

The vertical distribution of wildfire emissions, commonly referred to as plume height, is a key factor in determining their atmospheric impact. Plume rise is governed by fire intensity, heat flux, and atmospheric stability (Val Martin et al., 2012). While most emissions remain within the planetary boundary layer (PBL), a significant fraction (4–20% over North America) reaches the free troposphere, enabling long-range transport and interaction with cloud systems (Kahn et al., 2007; Val Martin et al., 2010).

Extreme wildfires can generate pyroconvective clouds, including pyrocumulus (pyroCu) and pyrocumulonimbus (pyroCb), which play a crucial role in determining plume height. These cloud types can inject smoke into the upper troposphere and lower stratosphere (UTLS), where aerosols may persist for weeks to months, thereby influencing the global radiation budget (Fromm

50

60

et al., 2022). This underscores the necessity of accurately modeling plume rise including the triggering of pyro-convective clouds, as it is essential for assessing the global climate impacts of wildfires.

Various techniques have been developed over the years to parameterize the injection height of wildfire emissions within atmospheric and chemical transport models. One simple approach is the use of a fixed emission height, as demonstrated by Colarco et al. (2004); Dirksen et al. (2009) and Lamarque et al. (2003). Dirksen et al. (2009) evaluated three fixed emission heights for intense forest fires in southeastern Australia (December 2006) and found that ground-level and 5 km emissions underestimated plume heights compared to CALIPSO observations, while emissions near 10 km aligned best with the satellite data.

Another approach involves prescribing an emission profile. Lavoué et al. (2000); Pfister et al. (2006) and Wang et al. (2006) used a uniform profile throughout the troposphere, while other studies, such as Generoso et al. (2007); Hyer et al. (2007) and Leung et al. (2007), emitted a pre-selected fraction of aerosol mass above the boundary layer. However, these assumptions often neglect the impacts of sensible and latent heat release by the fire and cloud microphysical processes and therefore emission profiles must be adapted for each case, as every fire exhibits unique characteristics.

Additionally, there are empirical models that parameterize the sub-grid processes of the fire-atmosphere interaction and return an emission height (Briggs, 1975; Lavoué et al., 2000; Sofiev et al., 2012). These approaches are computationally efficient and suitable for large-scale applications or operational forecasting, where detailed fire-atmosphere coupling is not feasible. However, they rely on simplified assumptions and static input parameters, which limit their accuracy under dynamic fire conditions.

For instance, the widely used Briggs parameterization was originally developed for industrial stack plumes and has been adapted for wildfire applications. While it provides a fast and practical method for estimating plume rise, it exhibits several key limitations. As noted by Raffuse et al. (2012), the model depends on static diurnal profiles of atmospheric stability and wind speed, which may not reflect rapidly evolving meteorological conditions during fire events. It also lacks representation of latent heat release and complex plume–atmosphere interactions. This can lead to an overestimation of plume heights, (Raffuse et al., 2012), particularly when compared to satellite observations.

A more advanced approach is the physical parameterization of the injection height, as developed by Freitas et al. (2006). The Freitas model is one of the most widely used plume-rise models embedded in numerical weather prediction (NWP) and global transport models. Val Martin et al. (2012) evaluated the Freitas plume-rise model using various estimates of active fire area and sensible heat flux, finding that the model often underestimated the dynamic range of plume heights and failed to consistently identify injections into the free troposphere. Several factors contribute to these limitations. First, the model is sensitive to uncertainties in input parameters, which is difficult to constrain due to the heterogeneous nature of fire behavior. Second, the assumption that the sensible heat flux constitutes 55% of the total heat flux may not hold universally, especially given the uncertainties in fire radiative power (FRP) retrievals from satellite observations such as MODIS. Third, the model relies on coarse-resolution meteorological data, which can misrepresent key atmospheric variables like stability and boundary layer height. Finally, limitations in the entrainment parameterization may lead to poor agreement with observed injection heights.

In addition to these performance issues, physically based models are computationally expensive.

To address these challenges, recent studies have explored machine learning (ML) as a promising alternative. For instance, Wang (2024) developed an ML-based plume rise emulator that demonstrated improved accuracy and significantly reduced computational cost compared to the Freitas model. Building on this, Lu et al. (2024) integrated an interactive fire plume-rise scheme into the Energy Exascale Earth System Model (E3SM), highlighting the potential of dynamic, data-driven modeling to enhance assessments of aerosol radiative effects.

Despite recent advances, ML approaches face significant limitations in contributing to the physical understanding of plume dynamics. While they may enhance predictive performance, they often lack interpretability and fail to capture the underlying physical processes driving plume behavior. One process that drives plume behavior is the diurnal variability of fire activity, which plays a critical role in determining plume rise characteristics (Walter et al., 2016). For instance, Ke et al. (2021) demonstrated that emissions during peak burning hours are more likely to be injected above the PBL. This finding is further supported by Li et al. (2023), who showed that accurate representation of plume-rise dynamics is essential for predicting surface-level PM<sub>2.5</sub> concentrations during extreme wildfire events.

Convection-resolving simulations have emphasized the dominant role of meteorological conditions in driving pyro-convection (Luderer et al., 2006). Importantly, fires themselves can modify local atmospheric conditions, creating feedbacks that are typically neglected in current plume-rise models. While this simplification may be justified at coarse spatial resolutions (30–100 km) (Freitas et al., 2007), it limits the accuracy of high-resolution simulations.

Beyond the initial injection height, aerosol radiative effects—particularly from absorbing aerosols, such as black carbon, can substantially influence plume evolution. Through the absorption of solar radiation, these aerosols heat the surrounding air, increasing buoyancy and promoting further vertical transport in a process known as aerosol-induced lofting (Muser et al., 2020). This mechanism can significantly modify the vertical distribution of smoke, extending its atmospheric residence time and enhancing long-range transport. As a result, aerosols lofted to higher altitudes can exert more persistent and widespread impacts on atmospheric composition and radiative forcing (Ohneiser et al., 2023).

Ohneiser et al. (2023) found that the lofting of wildfire smoke into the troposphere and stratosphere is highly sensitive to factors such as aerosol optical thickness (AOT) and meteorological parameters. Their simulations, validated against space lidar observations, highlighted significant lofting processes during pyroconvection events in Canada and Australia. Ma et al. (2024) showed that aerosol transport to the middle stratosphere during Australian New Year's event (ANY) occurred in three phases: pyro-convection up to 16 km and self-lofting due to radiative effects up to 25 km. The final ascent to around 35 km was driven by a stratospheric circulation vortex caused by aerosol absorption. Further, Heinold et al. (2022) examined the emissions from the Australian wildfires and highlighted that global models describing aerosol-climate impacts lack adequate descriptions of the emission height of aerosols from intense wildfires. Using a combination of aerosol-climate modeling and lidar observations, they demonstrate the importance of accurately representing these high-altitude fire smoke layers for estimating the atmospheric energy budget. The impacts of the direct and indirect aerosol effect are shown by Savenets et al. (2024), who outlines that simulations of direct, indirect, and combined aerosol effects led to colder and drier surface conditions. Changes in cloudiness, precipitation, and wind speed were observed. Larger uncertainties were noted in coarser models. Quantifying

105

125

aerosol effects is crucial for predicting and mitigating adverse weather conditions and wildfires, thereby improving emergency response measures.

The simulation of aerosol radiative effects critically depends on the assumed optical properties of the particles. These properties, such as single scattering albedo (SSA), extinction coefficient, and asymmetry parameter, are not constant but vary with several key factors. Most notably, they depend on particle size, which influences the scattering and absorption efficiency relative to the wavelength of incoming radiation (Chen et al., 2006). Morphology also plays a significant role, particularly for non-spherical or fractal-like structures such as fresh soot aggregates, which alter the angular distribution of scattered light (Pang et al., 2023). Additionally, the chemical composition determines the complex refractive index, which governs the intrinsic absorptive and scattering behavior of the particles (Bond and Bergstrom, 2006). The mixing state, whether particles are internally or externally mixed, further modifies their effective optical properties, especially in the presence of absorbing cores and non-absorbing coatings (Bond et al., 2006). Finally, hygroscopic growth under varying humidity conditions dynamically alters both particle size and refractive index (Petzold et al., 2005). Simplified assumptions, such as treating aerosols as spherical, internally mixed particles with fixed optical properties, may therefore introduce substantial uncertainties in the estimation of aerosol radiative forcing and its feedback on plume development. As shown by Brown et al. (2021), biomass burning aerosols in most climate models are too absorbing, leading to an overestimation of their warming effect.

This study focuses on the Australian New Year's fire event and aims to address limitations in current plume-rise models by explicitly incorporating the fire's impact on atmospheric conditions in the host model. In previous studies, the host model provided atmospheric conditions to the plume rise model, which in turn returned an injection height, as illustrated in Figure 1. Although the fire's impact is considered within the plume rise model, the host model itself remains unaware of the fire-induced changes to meteorological variables.

This study integrates fire-induced heat and moisture release, along with aerosol-radiation interactions for internally mixed aerosols, to investigate their combined influence on plume dynamics and vertical aerosol distribution. Understanding these coupled processes is essential for accurately assessing the role of wildfire emissions in atmospheric composition and their broader climatic impacts.

### 2 Methodology

## 120 2.1 ICON-ART modeling system

The model employed in this study is the ICOsahedral Nonhydrostatic (ICON) numerical weather and climate model. This model solves the fully three-dimensional, non-hydrostatic, and compressible Navier-Stokes equations on an icosahedral grid (Zängl et al., 2015). The ICON model facilitates seamless simulations of various processes across local to global scales (Heinze et al., 2017; Giorgetta et al., 2018). Additionally, the ART (Aerosol and Reactive Trace gases) module is activated. This module encompasses the emission, transport, physicochemical transformation, and removal of aerosols and trace gases (Rieger et al., 2015). The ART module includes detailed representations of aerosol microphysics, such as nucleation, coagulation, and condensation processes (Muser et al., 2020). It also accounts for aerosol-cloud interactions and the direct and indirect radiative

145

effects of aerosols. Comprehensive descriptions are provided in Rieger et al. (2015); Schröter et al. (2018) and Muser et al. (2020). ICON-ART employs the ecRad model by Hogan and Bozzo (2018), integrated into ICON, to handle radiative transfer calculations. The optical properties of cloud particles, aerosols, and gases vary with wavelength. ecRad covers a spectral range from 0.2 to 1000  $\mu$ m, divided into 30 spectral bands. These bands are used for aerosol and cloud particle optical properties, while the gas optical properties are further subdivided. Only upwelling and downwelling radiation are considered, and the optical properties are integrated over all angles, simplifying the necessary parameters to the mass extinction coefficient ( $k_{ext}$ ), the single scattering albedo ( $\omega$ ), and the asymmetry parameter (g). For aerosols, optical properties are supplied directly to the radiation scheme. Using this information, ecRad calculates reflection, transmission, and internal radiation sources for each grid box and model level, resulting in the upward and downward radiative fluxes. In addition, it calculates radiative heating and cooling, which feeds back into the dynamics and physics of the model.

#### 2.1.1 Wildfire emissions in ICON-ART

To address vegetation fire emissions, the ICON-ART modeling system is extended by incorporating the one-dimensional Freitas model (Freitas et al., 2006, 2007, 2010). It is implemented in ICON-ART in the same way as it was implemented in COSMO-ART by Walter et al. (2016). This coupling is briefly explained below.

The plume-rise model takes into account buoyancy, atmospheric stratification, and flow conditions to calculate the plume height. These processes occur on scales significantly smaller than the grid spacing of regional and global modeling systems, typically on the order of 100 meters for plume dynamic processes compared to 10-100 kilometers in the host model, such as ICON-ART.

The one-dimensional plume-rise model employs an internal vertical grid spacing of 100 m with 200 vertical layers. Environmental conditions, such as pressure, humidity, temperature, and wind speed are calculated by ICON-ART. For every grid point with an active fire, these variables are transferred to the plume-rise model to calculate the current plume height.

Fire size and intensity, based on vegetation type and density, determine heat release and initial buoyancy. The lower boundary condition assumes a virtual buoyancy source below the surface, resulting in high vertical velocity at the surface. Final buoyancy is limited by turbulent and dynamic entrainment, with turbulent entrainment causing dilution and increased plume radius, and dynamic entrainment accounting for wind speed and plume bend-over. Buoyancy is further increased by latent heat release during condensation. The plume top is defined as the height at which the vertical velocity drops below  $1 \text{ m s}^{-1}$ .

To determine the bounds for the effective emission height in the ICON-ART model, an upper and lower limit for the fire's heat flux is assumed. These heat flux values are dependent on the type of vegetation in the respective grid cell and are taken form Freitas et al. (2006).

In Walter et al. (2016), the fire area was set to 50 ha, this assumption is replaced by the approach of Val Martin et al. (2012), which is also used by Ke et al. (2021). Here, the fire size in a grid cell  $A_{ac}$  is given by:

$$A_{gc} = \Delta r \frac{FRP_{gc}}{FRP_{max}} \tag{1}$$

 $\Delta r$  is the horizontal resolution of detected fire dataset. For this application the Global Fire Assimilation System (GFAS) is used as fire input. GFAS has a resolution of  $0.1^{\circ}$ .  $FRP_{max}$  is the maximum FRP and defined as the 99th percentile value of the detected FRP in the fire region.  $FRP_{gc}$  is the fire radiative power in the respective grid cell.

The FRP from GFAS is used for the weighting of the fire size. Since aerosol, moisture, and heat release are also based on GFAS data, a brief description follows.

GFAS relies on FRP, which is derived from the NASA fire product MOD14. MOD14 includes thermal radiation observations ( $\lambda$ =3.9  $\mu$ m-11  $\mu$ m) from the MODIS (Moderate Resolution Imaging Spectroradiometer) instrument aboard the Terra and Aqua satellites (Giglio, 2007; Justice et al., 2011). Thermal radiation cannot penetrate clouds, making satellite observations of active fires reliable only in cloud-free regions. To address the data gaps caused by this limitation, fire data is assimilated using a Kalman filter (Rodgers, 2000). It is assumed that the true FRP density at time step (t) is a combination of the FRP density from the previous time step (t-t) and the observed FRP density at time step (t). Sampling is limited to a maximum of four times per day, and it is assumed that these three to four daily overpasses adequately represent the diurnal cycle of the fire (Kaiser et al., 2012). FRP measures the radiative energy released by the fire, which is assumed to correlate with the amount of vegetation burned and is proportional to biomass burning emissions (Kaiser et al., 2012).

The diurnal cycles of fire can be observed by geostationary satellites. The diurnal cycle function proposed by Kaiser et al. (2009); Andela et al. (2015) and applied by Walter et al. (2016) in COSMO-ART is also implemented in ICON-ART. To account for the diurnal variability, a diurnal cycle function  $d(t_1)$  is applied to the fire intensity and fire size.

$$d(t_1) = \omega + (1 - \omega) \frac{1}{\sigma \sqrt{2\pi}} \exp\left(-\frac{1}{2} \left(\frac{t_1 - t_0}{\sigma}\right)^2\right) \tag{2}$$

Here  $\omega$  is a weighting, which is set according to the vegetation type.  $\omega$  is 0.039 for tropical forests, 0.018 for savannas, and 0.003 for grasslands.  $t_1$  is the local solar time,  $t_0$  is the expected value of maximum emission set to 12.5 and  $\sigma$  is the standard deviation, set to 2.5. The diurnal cycle function is also applied to the emission fluxes.

The Plume-rise model returns both the plume bottom and top height. Following Walter et al. (2016), a parabolic emission profile ( $f(z^*)$ ) is assumed between these heights to represent the vertical distribution of emissions:

$$f(z^*) = 6z^*(1-z^*) \tag{3}$$

The dimensionless height  $z^*$  is defined as:

$$z^* = \frac{z - z_{bot}}{z_{top} - z_{bot}}$$
 (4)

z is the model height,  $z_{top}$  and  $z_{bot}$  are the plume top and bottom heights, respectively.

This leads to an emission rate E in kg m<sup>-2</sup> s<sup>-1</sup>, which is calculated for a respective grid cell and is depending on the height z and the time t.

$$E(z,t) = M(t) \times d(t) \times f(z,t) \times 3.4 \tag{5}$$

M is the daily mean emission flux in kg m<sup>-2</sup> s<sup>-1</sup> based on the GFAS dataset. For this study the aerosol emission flux is assumed to be the sum of the GFAS BC plus OC. d is the diurnal cycle given by equation 2 and f is the parabolic emission

Figure 1. Schematic illustration of the plume-rise model coupling in ICON-ART

profile between the upper and lower injection heights. To account for a systematical underestimation of the particulate matter emission in the GFAS dataset, the emission flux is multiplied by an empirical factor of 3.4 as suggested by Kaiser et al. (2012). Later comparisons will demonstrates that the application of this correction factor results in a generally good agreement with observed air quality measurements.

### 2.1.2 Heat and Moisture Release




Figure 1 illustrates how the ICON model provides atmospheric input to the plume-rise model. However, the effects of fire-induced heat and moisture release, as well as aerosol–radiative interactions, are not reflected in the atmospheric state used for plume-rise calculations. To address this limitation, the parameterizations for heat and moisture release proposed by Muth et al. (2025) are applied and adapted from a convection-resolving to a convection-parameterizing model setup. A brief description of this approach is provided below.

The total energy released by fires is calculated by multiplying the Fire Radiative Power (FRP) by a factor of 10, as proposed by Val Martin et al. (2012) and applied in Ke et al. (2021). To estimate the portion of this energy contributing to convective processes, a factor of 0.55 is used, following Freitas et al. (2006). Additionally, the FRP is weighted with a diurnal cycle function to account for peak fire intensity typically occurring in the early afternoon. The resulting heat release is implemented as a sensible heat flux from the surface to the atmosphere, leading to a fire-induced sensible heat flux  $sh_{fire}$  defined as:

$$sh_{fire} = FRP \times 5.5 \times d \times 3.4 \tag{6}$$







The FRP and  $sh_{fire}$  both have units of W m<sup>-2</sup>. We decided to apply the GFAS correction factor of 3.4 to the heat release estimates as well. Although this factor was originally validated for aerosol emissions only, we extend its application to FRP more broadly. We hypothesize that this adjustment helps compensate for potential underestimations of FRP in cases of intense pyroconvective activity and dense aerosol clouds, which may obscure satellite observations and result in lower FRP retrievals. The moisture release implementation includes combustion moisture with an emission ratio of 0.75 H<sub>2</sub>O/(CO+CO<sub>2</sub>) according to Parmar et al. (2008). CO and CO<sub>2</sub> emission fluxes from GFAS are scaled and incorporated into the ICON specific humidity tracer. Fuel moisture, divided into 30% dead and 70% alive components, follows thresholds from Nolan et al. (2016) and Deb et al. (2020), resulting in an approximate fuel moisture content of 75.42%, which is then multiplied by the GFAS combustion rate.

$$qv_{fire} = (0.75 \times (m_{CO} + m_{CO_2}) + 0.7542 \times m_{load}) \times d \times 3.4 \tag{7}$$

Moisture emission flux by the fire is  $qv_{fire}$  in kg m<sup>-2</sup> s<sup>-1</sup>.  $m_{CO}$  and  $m_{CO_2}$  in kg m<sup>-2</sup> s<sup>-1</sup> are the mass fluxes of CO and CO<sub>2</sub> and  $m_{load}$  in kg m<sup>-2</sup> s<sup>-1</sup> is the combustion rate. The emitted mass is weighted with a diurnal cycle function and again multiplied with the correction factor of 3.4. The moisture is added to the specific humidity tracer. The moisture emission follows the same injection height and profile as the particles, according to the plume-rise model.

#### 2.1.3 Aerosol Optical Properties

As previously noted, ICON-ART incorporates the aging processes of aerosol particles, which significantly influence their optical properties. The methodology employed to represent and simulate this aging is detailed in the following section.

The essential input variables for the ICON-ART radiation calculations include  $k_{ext}$ ,  $\omega$ , and g. These parameters are derived using a Mie code developed by Bond et al. (2006) and Mätzler (2002), based on the work of Bohren and Huffman (2008). Research by Brito et al. (2014) indicates that biomass burning aerosols develop a shell, leading to internal mixing. To account for this particle coating, a core-shell model can be incorporated into the calculations. The input parameters required for the Mie calculations are the median diameter, the shell-to-core fraction, and the refractive index of core and shell. For this study a bi-modal aerosol size distribution with a median number diameter of  $d_n = 20$  nm for the smaller Aitken mode and  $d_n = 150$  nm for the larger accumulation mode is assumed. The aerosol size distributions and chemical compositions used in this study are consistent with the ranges reported by Brito et al. (2014), based on ground-based measurements during the SAMBBA campaign in September 2012. These findings are further supported by Levin et al. (2010), who investigated the physical, chemical, and optical properties of biomass burning aerosols during controlled combustion experiments in the FLAME campaign. Additionally, Sakamoto et al. (2015) reported similar aerosol characteristics from in-flight sampling conducted with the FAAM research aircraft. Measurements reported by Brito et al. (2014) indicate soluble-to-insoluble fractions of 0.07 and 0.12 for two distinct observational periods. In this study, a shell-to-core volume fraction of 0.1 is adopted, consistent with the range reported by Reid et al. (1998b). Additionally, a BC/OC mass ratio of 0.03, which is within the range reported in Konovalov et al. (2017) is assumed and and inorganics/H<sub>2</sub>O mass mixing ratio of 0.75. The inorganics-to-H<sub>2</sub>O ratio is derived from an analysis of

Figure 2. Optical properties of OC+BC containing aerosol modes at ecRad wavelengths. a) mass extinction  $k_{ext}$  in m<sup>2</sup> g<sup>-2</sup>, b) single scattering albedo  $\omega$ , and c) asymmetry parameter g unit-less. The black lines show the Aitken mode and the brown lines the accumulation mode. Solid and dashed lines show the insoluble (uncoated aerosol) and mixed (coated aerosol) mode, respectively.

the reference experiment, acknowledging that this parameter is highly uncertain and subject to temporal variability (Zauscher et al., 2013). The results of the Mie calculations is shown in Figure 2. It is evident that the mixed modes (coated aerosol) exhibit slightly higher extinction coefficients and single scattering albedo in the visible range compared to the insoluble modes (uncoated aerosol). This is attributed to the H<sub>2</sub>O–H<sub>2</sub>SO<sub>4</sub> coating, which acts as a strong scatterer. The Mie theory assumes that particles are spherical. In reality, soot and biomass burning particles have a variety of morphologies: chain aggregates, solid irregulars and more liquid/spherical shapes. The physical and chemical composition is strongly variable and depending on the fuel type, moisture, combustion phase, wind conditions and age of the particles (Reid et al., 1998a). However, the liquid coating can lead to spherical particle surfaces, justifying the assumption of particle sphericity in the mixed mode. For consistency, the sphericity assumption is also applied to the insoluble mode containing uncoated particles, as it is done in Muser et al. (2020).

## 2.2 Observational data




As a case study, the Australian Black Summer Fires 2019/2020, particularly the initial phase of the Australian New Year's event, has been selected to test the developments. The interaction between the fire, the plumes and the atmosphere during this event was significant. Record-breaking warmth in December, coupled with exceptionally low rainfall, created conditions conducive to extreme fire activity. The ANY event was especially significant due to the passage of a cold front through southeastern Australia. This front was preceded by elevated temperatures and strong wind speeds, which intensified fire behavior. During





this event, the fires generated 38 pyroCb clouds, during 18 sub-events (Peterson et al., 2021). This type of intense pyroconvection represents an extreme case and may not be ideal for generalizing fire—atmosphere interactions under typical conditions. However, it provides an excellent opportunity to investigate how extreme wildfires influence meteorological variables and drive plume development.

To validate the results the NASA 3D wind algorithm and CALIPSO data is used. The NASA 3D wind retrieval algorithm, as described by (Carr et al., 2018, 2019, 2020), is utilized to determine the heights of plumes and clouds. This algorithm employs stereo imaging, which uses geometric parallax to derive feature heights. By integrating data from geostationary (GEO) and low-earth orbit (LEO) satellites, it produces three-dimensional (3D) atmospheric motion vectors (AMVs) through a multiplatform, multi-angle stereoscopic approach. The term "3D Winds" refers to the three-dimensional positioning of horizontal AMVs within the atmosphere. Observing the parallax of a feature from two different vantage points (stereo) provides direct information about its height.

In this study, the LEO-GEO retrieval method is applied. The LEO satellite data is sourced from Terra and Aqua MODIS Level 1B in the blue band (459-479 nm) with a 500 m resolution. The GEO satellite data is obtained from Himawari-8's blue band (430-480 nm). The Advanced Himawari Imager (AHI), operated by the Japan Meteorological Agency, has a 10-minute temporal resolution that is used to track feature movement. MODIS data is then used to calculate parallax, determining AMVs and height. A quality flag is employed to exclude poor retrievals.

Further, analysis of CALIPSO (Cloud-Aerosol Lidar and Infrared Pathfinder Satellite Observation) data is used for validation. The CALIPSO satellite integrates an active lidar instrument with passive infrared and visible imagery to analyze the vertical structure of thin clouds and aerosol layers globally. Launched in 2006 alongside the CloudSat satellite's cloud profiling radar system, CALIPSO provided valuable data on atmospheric clouds and aerosols. The data utilized in this study is the total attenuated backscatter at 532 nm, classified as Level 1 data. These attenuated backscatter profiles are derived from the calibrated, range-corrected, laser energy normalized, and baseline-subtracted lidar return signal. The horizontal resolution ranges from 0.33 km to 5 km, while the vertical resolution varies from 30 to 300 m, depending on altitude.

Finally, air quality data from the Centre for Air Pollution, Energy and Health Research's (CAR) National Air Pollution Monitor Database (NAPMD) were utilized (CAR's NAPMD, 2021). To reduce noise in the data, three-hourly mean concentrations of PM<sub>2.5</sub> are calculated and compared with corresponding model-derived aerosol concentrations averaged over the respective grid cells. The locations of the monitoring stations are shown in Figure 3.

# 2.3 Model Configuration

Limited-area model simulations are conducted with a grid spacing of 6.6 km, employing parameterized convection and incorporating a plume-rise model to represent injection heights. It is proposed that at this resolution, the influence of fires on meteorology becomes significant and should not be neglected. However, the grid spacing is still too coarse to explicitly resolve convection and the associated plume-rise processes. Prior to the experimental simulations, a global simulation with a grid spacing of 13 km is performed to obtain the input data for the boundary conditions. This global simulation is initialized using the German Weather Service (DWD) analysis product and does not account for fire impacts on the meteorological variables. The


**Figure 3.** The simulated domain including the GFAS FRP (CAMS, 2021) remapped to the ICON grid on the December 30, 2019. Generated using Copernicus Atmosphere Monitoring Service information [2023]. The magenta markers show the locations of the air quality stations. The green line shows the CALIPSO overpass on January 2, 2020, at 02:30 AEDT, the blue line corresponds to the CALIPSO overpass on January 1, 2020, at 13:30 AEDT

experiment domain extends from southeast Australia to New Zealand, as shown in Figure 3. The experiments are initialized with the DWD analysis product and meteorological variables of the boundary conditions are read every 3 hours. The domain is 30 km high with 70 vertical levels, the vertical grid spacing increases with height. Simulations run from December 30, 2019, at 00:00 UTC to January 1, 2020, at 22:00 UTC and therefore focus on the first phase of the ANY event.

Chemical tracers (CH<sub>4</sub>,  $C_2H_6$ ,  $C_3H_8$ , CH<sub>3</sub>COCH<sub>3</sub>, CO, NH<sub>3</sub>, NO<sub>2</sub>, SO<sub>2</sub>, DMS, HNO<sub>3</sub>) are initialized using CAM-Chem data (Buchholz et al., 2019; Emmons et al., 2020). A simplified OH-chemistry mechanism (Weimer et al., 2017) is employed, which additionally includes  $C_5H_8$ , CO<sub>2</sub>, and OCS. The simulation allows for new particle formation through nucleation, particle coagulation, and condensation of gaseous species. ISORROPIA II (Fountoukis and Nenes, 2007) is used for gas-to-particle partitioning, accounting for  $H_2O$ , NO<sub>3</sub>, and NH<sub>4</sub>.

Fire emission data from GFAS is updated daily at 00:00 UTC. The FRP from GFAS is displayed for December 30 in Figure 3. The GFAS emission flux is proportional to the FRP, indicating fire locations and emission strength. On December 30, the FRP



**Table 1.** Overview of the performed experiments.

| Experiment | Moisture release | Heat release | Aerosol-radiation interaction |
|------------|------------------|--------------|-------------------------------|
| REF        | ×                | ×            | ×                             |
| MOIST      | ✓                | ×            | ×                             |
| HEAT       | ×                | ✓            | ×                             |
| ARI        | ×                | ×            | ✓                             |
| ALL        | ✓                | ✓            | ✓                             |

reaches up to  $144 \text{ W m}^{-2}$ , on December 31 up to 47 W m<sup>-2</sup>, and on January 1 up to 10 W m<sup>-2</sup>. This indicates a decrease of fire activity throughout the simulation.

In this study, aerosols are treated as an internal mixture of BC and OC to their co-emission from wildfire sources. This representation assumes that both components are part of the same aerosol particle, rather than existing as separate particles. It captures the fact that BC and OC are emitted together and are physically mixed during and shortly after emission. This approach provides a simplified but realistic way to represent wildfire aerosol composition in transport models. 6% of the particles are emitted in the smaller Aitken mode, with a log-normal distribution around the median diameter of 20 nm and standard deviation of 1.7, and 94% in the accumulation mode with a with a log-normal distribution around the median diameter of 70 nm and standard deviation of 2.0. The simulations do not account for aerosol-cloud interaction.

We performed five experiments (Table 1): a reference run (REF) without fire impact on meteorology, followed by three individual experiments isolating the effects of fire-induced heat release (HEAT), moisture release (MOIST), and aerosol-radiation interaction (ARI), and finally one experiment combining all three processes (ALL).

#### 2.4 Definitions and analysis methods

In the following, the temporal evolution of the mass-weighted plume height is presented. For this analysis, the plume is defined as the set of grid cells in which the mass mixing ratio exceeds 0.05  $\mu$ g m<sup>-3</sup>. The mass-weighted height is calculated by summing the product of the aerosol mass in each plume grid cell and the corresponding height of the grid cell center. This total is then divided by the overall plume mass to obtain the mass-weighted height.

An additional analysis of surface-based Convective Inhibition (CIN) and Convective Available Potential Energy (CAPE) was performed. Table 2 presents the mean CIN and CAPE values over the fire area derived from the experimental scenarios. The fire area is defined as a grid cell with an aerosol emission larger than  $5 \times 10^{-12}$  kg m<sup>-2</sup> s<sup>-1</sup>, taken from the GFAS dataset. CIN, expressed in J kg<sup>-1</sup>, quantifies the energy required to lift a surface air parcel to its level of free convection, thereby representing a measure of atmospheric stability. CAPE, also in J kg<sup>-1</sup>, denotes the amount of buoyant energy available to support convective updrafts, serving as an indicator of the potential intensity of convective processes.

For comparison with the NASA 3D wind algorithm, the top height includes contributions from both aerosol plumes and cloud layers, as the retrieval algorithm is unable to distinguish between the two. Therefore, an aerosol plume is defined as

**Figure 4.** Temporal evolution of mass weighted height of the plume during the first two simulation days. REF is in black, MOIST in blue, HEAT in red, ARI in green, and ALL in purple. The gray dotted lines indicate 14:30 AEDT on the 30 December and the 31 December.

a grid cell with an aerosol AOD at 550 nm divided by the vertical extend of the grid cell exceeds a threshold of  $50 \times 10^{-6}$ . A cloud is defined as a grid cell in which the combined sum of Liquid Water Content (LWC) and Ice Water Content (IWC) exceeds  $0.01 \times 10^{-3}$  g m<sup>-3</sup>. The top height is then determined as the highest altitude level at which either of these thresholds is exceeded. The comparison between CALIPSO attenuated backscatter and the simulated backscatter reveals several differences. First, the model provides only the total backscatter, not the attenuated backscatter. Second, the simulated backscatter accounts exclusively for wildfire aerosols, excluding contributions from other aerosol sources and clouds. To address the latter limitation, an isosurface corresponding to a combined LWC+IWC of  $01 \times 10^{-3}$  g m<sup>-3</sup> included in the plots.

## 3 Results





#### 3.1 Impact of Fire-Atmosphere Interaction on Plume and Clouds

In the subsequent analysis, the impact of fire-atmosphere interactions on plume height and cloud development is examined. The results are presented for the simulated time given as AEDT (Australian Eastern Daylight Time), which corresponds to UTC+11 hours. Figure 4 illustrates the plume mass-weighted height for the performed experiments. In the REF experiment, the mass-weighted height increases to a peak at 3.8 km after 4 hours, and declines by 1.2 km within the next 48 hours. Since there is no effect of the fire on the background meteorology, evolution during the first day is caused by the destabilization of the atmosphere, which is used to calculate the emission height in the plume rise mode and the diurnal cycle of the fire within the plume-rise model. The increasing fire intensity and instability lift the emitted aerosol masses. As previously explained, FRP and consequently the emission strength decreases over the time span of the simulation. Therefore, the mass-weighted plume height is primarily influenced by emissions occurring during the first day of the simulation. The decline observed after the peak can be attributed to sedimentation processes and emissions occurring at lower altitudes. The MOIST experiment follows a similar evolution as REF, but with a lower peak at 3.5 km after 3.5 hours. The impact of water vapor release is


**Figure 5.** Mean vertical profile within the fire area at 14:30 AEDT on December 30. a) Aerosol concentration. b) LWC (solid line) and IWC (dashed line). The profiles are color-coded as follows: REF (black), MOIST (blue), HEAT (red), ARI (light green), and ALL (purple).

greatest on the first day, when emissions are strongest, and has no significant impact on the overall mass weighted height development thereafter. In contrast, the HEAT experiment shows substantially higher values, peaking at 6.3 km after 4.5 hours, and minimum values of 4.0 km after two days. This can be explained by two main factors. First, the sensible heat release by the fire destabilizes the atmosphere and creates buoyancy, which lifts the plume higher and second, this less stable atmosphere is now used by the plume-rise model to calculate the injection height, thereby increasing the height. The effect is strongest on the first day, as it is proportional to the FRP. Constant emissions at lower levels and sedimentation decrease the mass-weighted height thereafter. The ARI experiment starts with the lowest mass-weighted height. This can be explained by the absorption of solar radiation by the dense aerosol plume above the fire area. This leads to a stabilization of the atmosphere over the fire area, which reduces the calculated injection height by the plume-rise model. The maximum height of 4.3 km is observed on December 31st at 20:30 AEDT. The scattering and mainly absorption of solar radiation can warm the plume and create buoyancy. The ALL experiment exhibits the most pronounced plume development, the top height of 7.1 km is reached after 5 hours, followed by a decrease during night and an increase after noon. This shows that destabilization due to the sensible heat release counteracts the stabilizing effects of additional cloud formation and aerosol-radiation interaction. It appears that the lofting effect in comparison to ARI is increased.








To explain the initial difference in mass-weighted height Figure 5 illustrates the mean vertical profile of the mean aerosol concentration over the fire area in a) and LWC and IWC over fire area in b), on December 30 at 15:00, two hours after peak fire intensity in the simulation. In Figure 5a, the REF experiment shows two peaks in the aerosol concentration: the maximum at 4.7 km and a local peak at 0.7 km. The mean LWC and IWC in Figure 5b shows, that in the REF experiment clouds form between 3.5 km and 14.4 km, with a peak condensate at 5.4 km. This indicates some convection, at least in parts of the fire area, which can further be connected to the passing cold front and the thereby caused instability. The aerosol concentration in the MOIST experiment also shows two peaks, however the upper peak is reduced by 31.6 %, while the lower peak increases by 39.3 %. IWC and LWC also show a similar distribution as REF. However, the peak of LWC increases by 30 %. It is shown that additional water vapor leads to more cloud formation in areas of preexisting clouds. The additional energy from the latent heat release, which is hypothesized to lift the aerosols higher, is counteracted by the stabilization of the atmosphere due to cloud radiative effects. This more stable atmosphere is now used by the plume-rise model to calculate the emission height. The aerosol concentration in the ARI experiment again shows two peaks. In comparison to REF and MOIST, the maximum is closer to the surface at 0.2 km height with a mean concentration more than twice as high as the REF maximum. Further, in the ARI experiment, there is an overall decrease in LWC, with a reduced mean of 50 % at 5.4 km. This reduction indicates a stabilization of the atmosphere due to aerosol-radiation interaction, and results in most of the aerosol plume remaining close to the surface. The IWC remains comparable to the REF experiment. The aerosol distribution in the HEAT experiment exhibits a markedly different vertical profile, characterized by a more stratiform structure with emissions reaching up to 13.9 km. Notably, two smaller peaks at 4.7 km and 8.5 km are observed, with a maximum concentration of 121.5  $\mu$ g m<sup>-3</sup>. Additionally, cloud formation increases. The LWC rises to levels comparable to those in the MOIST experiment. A local maximum at 12.1 km further indicates enhanced cloud ice formation, suggesting that clouds can develop even in the absence of additional moisture. In the ALL experiment, the vertical aerosol distribution shows emissions extending up to 14.8 km. The concentration peaks at 5.0 km; however, due to higher overall emissions, this peak is reduced by 67% compared to the REF experiment. Nearsurface concentrations are only slightly reduced by up to 19% relative to REF. This highlights, on one hand, the increased buoyancy resulting from heat release and enhanced convection, and on the other hand, the stabilizing effects of aerosol- and cloud-radiation interactions. The ALL experiment integrates all contributing factors, producing the strongest cloud response. Although the peak at 5.5 km increases only slightly, the LWC shows a consistent rise up to 9.9 km. The IWC exhibits a distinct local maximum at 10.8 km, with peak concentrations increasing by a factor of 5.7 compared to REF. It is important to note that the presented profiles represent spatial averages; thus, the stabilizing effects are more pronounced in regions with smaller fires, while increased convective cloud formation is observed in areas with higher emissions and more intense fires. Furthermore, a considerable amount of aerosols is emitted above the cloud peak concentration level in the ALL experiment, whereas in the ARI experiment, the majority of aerosols remain below the cloud layer. This suggests that the lofting effect in the ALL experiment may be stronger due to less absorption and scattering of solar radiation by clouds above the aerosol layer, allowing more energy to reach and heat the aerosol layer, thereby enhancing vertical transport. Additionally, the influence of the different experimental setups on surface pollutant concentrations is noteworthy, given the sub-

stantial impact of wildfire emissions on air quality. The results suggest that the stabilizing effects of aerosol- and cloud-radiation



**Table 2.** Mean CIN and CAPE in the fire areas for the REF, MOIST, HEAT, ARI, and ALL experiments on December 30, 15:00 AEDT. The values are given J  $kg^{-1}$ .

|       | CIN [J kg <sup>-1</sup> ] | CAPE [J kg <sup>-1</sup> ] |
|-------|---------------------------|----------------------------|
| REF   | 14                        | 3653                       |
| MOIST | 18                        | 3577                       |
| HEAT  | 0                         | 3781                       |
| ARI   | 73                        | 3091                       |
| ALL   | 26                        | 3507                       |

interactions lead to elevated surface concentrations compared to scenarios dominated by heat release, indicating that the former are associated with more uniform emission profiles.

In the next step, the impact of the different experiments on atmospheric stability is examined. Table 2 indicates that all experiments exhibit atmospheric instability within the fire areas, with CAPE values suggesting the potential for deep convection. However, the interplay of fire-induced effects including moisture release, heat fluxes, and aerosol-radiation interactions leads to notable variations in both CIN and CAPE. These variations reflect the complex feedback mechanisms between surface emissions and convective processes. In the REF experiment, CIN is 14 J kg<sup>-1</sup> and CAPE is 3653 J kg<sup>-1</sup>, providing baseline conditions for convective development. The addition of water vapor in the MOIST experiment slightly increases CIN to 18 J kg<sup>-1</sup> and decreases CAPE to 3577 J kg<sup>-1</sup>. Increased humidity generally leads to more rapid saturation of rising air parcels, resulting in the release of latent heat. As a consequence, the parcel becomes positively buoyant, meaning it is more capable of rising through the atmosphere. Thus, CIN/CAPE should decrease/increase as humidity increases. However, Figure 5 also indicates an increase in cloud formation, which contributes to atmospheric stabilization through cloud radiative effects. This stabilization is primarily associated with surface cooling due to reduced incoming solar radiation. Nevertheless, the overall changes remain small. The ARI experiment exhibits the highest CIN (73 J kg<sup>-1</sup>) and the lowest CAPE (3091 J kg<sup>-1</sup>), consistent with aerosol-radiation interactions stabilizing the lower atmosphere and suppressing convective activity. In contrast, the HEAT experiment shows the strongest convective potential, with no inhibition (CIN = 0 J kg<sup>-1</sup>) and the highest CAPE (3781 J kg<sup>-1</sup>), suggesting that fire-induced heating strongly promotes convective initiation and vertical development. In the combined experiment ALL, CIN is 26 J kg<sup>-1</sup> and CAPE is 3507 J kg<sup>-1</sup>, reflecting the net effect of competing processes. Again, the values represent area-averaged quantities. The highest CIN values indicate the presence of regions with increased atmospheric stability, typically associated with smaller fires. In contrast, larger fires exhibit substantial heat release, which counteracts the stabilizing effects and promotes convective activity.

In a next step, the impact of aerosol-radiation interaction on the cloud development is analyzed. Figure 6a displays the sum of LWP + IWP for the reference experiment. The individual contributions of LWP and IWP are shown separately in Appendix A1 and A2. A cloud band spreads from the northwestern corner southeast through the domain, overlapping with the aerosol plume. This cloud band is associated with a front passing through. Figure 6b shows the difference of ARI - REF. There is noise within the cloudy areas, but significant increases in LWP + IWP are observed in the southeastern region of the cloud band.




Figure 6. LWP+IWP for the a) REF, difference in LWP+IWP for b) ARI-REF, and c) ALL-REF on December 31 at 14:30 AEDT.

This clear increase stems from an increase in liquid clouds forming due to the aerosol-radiative effect. West of the cloud band, there is an increase in cloud ice caused by aerosol-radiative effects. Figure 6c shows the differences between ALL and REF. Again, there is noise, but also areas of increase. The areas of increase are evident in the southeastern part of the cloud band and east of the cloud band where the plume is located. The increases within the cloud areas are predominantly in liquid clouds, while east of the cloud band, there is a dominant increase in cloud ice. Therefore, aerosol-radiation induced plume warming increases the buoyancy of rising air parcels, allowing the plume to ascend higher and promote cloud formation. In contrast to the beginning of the simulation, where the dense aerosol plume induces atmospheric stabilization. This highlights the semidirect aerosol effect, which can exert both stabilizing and destabilizing influences on atmospheric dynamics, depending on the vertical distribution and optical thickness of the aerosol plume. When aerosols absorb solar radiation, they can heat the surrounding air, potentially stabilizing the lower atmosphere by suppressing convection. This effect tends to trap pollutants near the surface, enhancing surface concentrations, as seen close to the source. Conversely, if the aerosol layer is elevated and sufficiently dense, it can destabilize the atmosphere by creating temperature gradients that promote vertical mixing. The net impact of the semi-direct effect is therefore highly sensitive to plume height, aerosol concentration, atmospheric stratification and the aerosol optical properties. Furthermore, a distinction between the ARI and ALL experiments is evident. While some of these differences can be attributed to initial conditions, such as the plume injection profile, the vertical positioning of the aerosol plume relative to cloud layers further contributes to the observed discrepancies.

#### 3.2 Comparison to observations

The following section presents a comparative analysis of the different model experiments against observational datasets, including the horizontal and vertical aerosol transport patterns derived from NASA 3D wind retrials and CALIPSO, and the resulting impacts of model modifications on surface-level PM<sub>2.5</sub> concentrations at selected monitoring stations.

Figure 7 shows the 3D wind retrieval compared to the five experiments on December 31st between 13:45 and 15:25 UTC and the simulation at 14:30 UTC. The retrieval in Figure 7a shows a feature almost diagonal through the domain which will be



**Figure 7.** Aerosol and cloud top height on December 31, 2019, at 14:30 UTC. a) Retrieved by the NASA 3D wind algorithm. b-f) Simulated in the experiments: b) MOIST, c) ALL, d) REF, e) ARI, and f) HEAT. The dots in panels b-f represent the top height of either the aerosol plume or a cloud.

focus of the analysis. Within that feature, there is a convective cell arising southeast of the Australian coastline, with maximum altitudes of 20.0 km. The average heights of the plume and cloud above 2 km are 8.3 km. The REF experiment exhibits similar structural features to those observed in the retrieval. It captures the overlap between the aerosol plume and cloud layers (LWP+IWP is shown in Figure 6 the plume top height can be found in Appendix A3). Within, there are two distinct elevated regions are evident: one shifted toward the southeast, characterized by both cloud and aerosol presence, and another toward the northwest, dominated primarily by aerosols. The maximum top height within the REF experiment reaches 13.0 km and the average height above 2 km are 6.7 km. This outlines an underestimation of the top heights in REF. The MOIST experiment exhibits similar characteristics to the REF experiment, but with more extensive elevated regions, outlined by an average top height of 7.0 km and an increase in maximum height of 13.4 km. While the maximum plume height is notably enhanced in the MOIST experiment, the average plume height shows only minor variation. These results suggest that the release of moisture influences the initial stages of plume development; however, its overall impact on the vertical structure remains limited. The ARI experiment also exhibits similarities to the REF experiment but demonstrates enhanced plume elevation downstream, with a maximum top height of 13.0 km and an average top height of 7.3 km. Aerosol-radiation interactions contribute to a gradual rise in plume altitude over time. Initially, the top heights in the ARI experiment are underestimated; however, as the plume ascends due to these interactions, the top heights increasingly align with observational data, showing improved agreement compared to the REF experiment. In the HEAT experiment, a pronounced elevation of the plume is observed downstream of the coastline, closely matching the spatial distribution seen in the observations. The maximum top height reaches 14.4


Figure 8. CALIPSO attenuated backscatter at 532 nm and simulated backscatter at 532 nm on January 1, 2020, at 13:30 AEDT. a) CALIPSO data. b-f) Simulated in the experiments: b) MOIST, c) ALL, d) REF, e) ARI, and f) HEAT. The light yellow line displays the  $0.01 \times 10^{-3}$  g m<sup>-3</sup> LWP+IWP isosurface

km, with an average top height of 7.6 km. Although an underestimation remains, the HEAT experiments aligns agreeably with the observations. In the ALL experiment, a dominant elevation is simulated southeast of the Australian coast, with a maximum altitude of 18.0 km and an average top height of 8.6 km, aligning even better with the observations. The maximum plume height is still underestimated; however, the retrievals reach altitudes of up to 20 km in localized regions. In contrast to this underestimation, the average top height exceeds the observed value by 0.5 km. Nevertheless, the initially higher and more uniform emission profile, combined with lofting induced by aerosol–radiation interactions in the ALL setup, effectively reproduces the observed patterns.

This comparison of ARI and ALL highlights that the aerosol radiative effect is strongly modulated by both cloud cover and the relative positioning of the aerosol plume. The initial lofting, driven by fire-induced heat and moisture release, increases the area of the plume exposed to solar radiation. This enhancement is attributed to the elevated initial plume height, its positioning above optically thick clouds, and a broader horizontal distribution.

The CALIPSO attenuated backscatter at 532 nm is compared to the simulated backscatter from the experiments (Figure 8), along a satellite overpass near New Zealand on January 1, 2020, at 13:30 AEDT. The CALIPSO cross-section reveals signal centered around 15 km altitude, accompanied by a secondary feature near 5 km in the southern portion of the transect, both classified as a mixture of cloud and aerosol. Additionally, a signal is visible above 10 km in the northern half of the overpass, classified as cloud. The surface close signal is classified as mainly marine aerosol with localized clouds. In the REF and MOIST experiments, no aerosol backscatter is simulated. However, the simulated  $01 \times 10^{-3}$  g m<sup>-3</sup> LWP+IWP isosurface around 10 km



Figure 9. CALIPSO attenuated backscatter at 532 nm and simulated backscatter at 532 nm on January 2, 2020, at 02:30 AEDT. a) CALIPSO data. b-f) Simulated in the experiments: b) MOIST, c) ALL, d) REF, e) ARI, and f) HEAT. The light yellow line displays the  $0.01 \times 10^{-3}$  g m<sup>-3</sup> LWP+IWP isosurface

aligns well with the observed cloud structures, and some low-level clouds are also simulated. The HEAT experiment captures the cloud signals and shows a small aerosol signal at the southern border of the shown overpass, indicating that the initially increased injection height improves aerosol transport compared to observations. The ARI experiment successfully reproduces the observed cloud signals. Additionally, an aerosol backscatter signal around 5 km is present at the southern edge of the overpass, consistent with CALIPSO observations. This suggests that a lofting mechanism is necessary and more effective than an increased emission height to transport aerosols to the observed location. The ALL experiment captures the cloud layer at 10 km but lacks near-surface cloud features. The aerosol backscatter matches the observations well, reproducing both the signal at 5 km and the elevated layer at 15 km. Overall, the findings highlight the limited impact of fire-induced moisture on aerosol transport in the MOIST experiment. The absence of a signal in the HEAT experiment suggests that, despite a higher initial emission profile, aerosol lofting is essential for transporting particles into the UTLS. Moreover, the signal around 5 km underscores the importance of lofting processes at lower altitudes. It should be noted that no clouds are simulated within the regions containing aerosol in the model output. Given reports of potential misclassifications of dense aerosol plumes in high altitudes (Liu et al., 2019), it remains unclear whether this discrepancy arises from limitations in the simulation or from flaws in the observational classification. Furthermore, the simulated backscatter includes only wildfire aerosols, hence, the nearsurface marine aerosol layer is not represented. The absence of low-level clouds in the ALL experiment further underscores the semi-direct aerosol effect, where stabilization trough aerosols-radiation interaction suppresses cloud formation.






Figure 9 presents the CALIPSO 532 nm attenuated backscatter cross-section south of the Australian coast on 2 January 2020 at approximately 02:30 AEDT. The satellite data reveals backscatter between 2 and 5 km, with additional features at 10 km (31°S), 8 km (34°S), and 12 km (between 43°S and 38°S), all classified as clouds by the CALIPSO algorithm. The model simulations also reproduce cloud structures in these regions, including those south of 43°S. The REF experiment aligns well with the observations, capturing the signals below 2 km north of 43°S. The signal north of 38°S matches well in both height and magnitude. The LWP+IWP isosurface matches, although shifted to the south with the detected cloud signal. The other experiments show similar results to REF and therefore also align well with the observations. This agreement indicates that the baseline model configuration is sufficient to reproduce the observed plume structure under these conditions. The similarity between REF and the other experiments (MOIST, ARI, HEAT, and ALL) suggests that fire-atmosphere interactions have a limited influence on plume height in this case.

Therefore, the differences between the two CALIPSO observations need to be discussed. Firstly, the age of the plume is different. In Figure 9, the plume is close to the source on January 1st. It has been established that on this day the FRP is smallest, resulting in lower moisture, heat release, and aerosol emission. Secondly, the overpass occurs during the night, when the atmosphere is more stable. This generally decreases vertical transport in all experiments. The diurnal cycle reduces heat and moisture during the night further, and the aerosol-radiative effects are limited to terrestrial radiation and remain small. All these factors reduce the impact of the implemented features, leading to similar results. However, the good agreement with the observations and the significant decrease in fire and moisture release indicate that for less intense fires, the plume-rise model performs well without the additional implementations. Therefore, it can be concluded that the fire's impact on the meteorology in the host model can be neglected for small to moderate fires. However, for extreme events, these effects are crucial.

So far the focus has been on the plume development and height but last we want to analyze how the different emission profiles shown in Figure 5 impact the surface-level air quality. Figure 10 compares simulated wildfire aerosol concentrations below 2.5  $\mu$ m in diameter with observed PM2.5 measurements at several monitoring stations (locations shown in Figure 3). It is important to note that, the measurements reflect total particulate matter, including non-wildfire sources not represented in the model. These factors introduce inherent uncertainties in the comparison. However the close proximity of the stations to fire areas (Figure 3) suggests that wildfire aerosols are the dominant source. At Station Albury, all simulations tend to underestimate PM2.5 concentrations; however, the observed peak on 31 January is reasonably well captured, except for the HEAT experiment. At Stations Bringelly and Campbelltown West, which are located near the emission source and in close proximity to one another, the REF and MOIST and HEAT experiments overestimate observed concentrations. In contrast, the ARI, and ALL experiments show improved agreement with measurements, indicating that the inclusion of fire-induced processes helps constrain near-source concentrations. At Station Newcastle, inter-experiment variability is minimal, with consistently lower concentrations in the HEAT, ARI, and ALL experiments. Overall, the simulations reproduce the observed concentration patterns reasonably well, thereby validating our assumptions regarding the aerosol emission flux, which consists of BC and OC, using a correction factor of 3.4.

Although Figure 5 indicates a strong increase in surface concentrations in the ARI experiment, this enhancement is not reflected in the comparison with air quality measurements. However, point-to-point comparisons remain sensitive to small discrepancies

**Figure 10.** Comparison of 3-hourly mean PM2.5 air quality measurements and simulated ICON-ART aerosol at four different locations, shown in figure 3. Observations are shown in gray, the simulation REF in black, MOIST in blue, HEAT in red, ARI in light green, and ALL in purple.

in simulated plume height and transport pathways, which can lead to substantial local deviations. This is particularly evident at Stations Bringelly and Campbelltown West, where, despite their close spatial proximity (approximately 20 km), the temporal evolution of concentrations differs markedly, underscoring the complexity of near-source plume dynamics In summary, the impact of varying experimental configurations appears to be non-linear. In some cases, enhancements such as heat release lead to a reduction in near-surface aerosol concentrations, whereas in others, concentrations increase. Notably, at the Bringelly and Campbelltown West stations, the overestimations observed in the REF, MOIST, and HEAT experiments are mitigated in the ARI and ALL setups, highlighting the importance of lofting for accurate near-source air quality representation.

#### 4 Discussion







First and foremost, the reasonability of our approach should be discussed. Different studies have outlined the limitations of the Freitas scheme (Val Martin et al., 2012; Wilmot et al., 2022; Wang, 2024), showing an underestimation of the dynamic range of emission heights. We showed, that this leads to an underestimation of plume heights for intense fires in comparison with observations. We addressed this by considering the fire's impact on meteorological variables. Therefore, we deliberately selected a configuration with a 6.6 km resolution, which does not explicitly resolve convection, in order to test the model's limitations. This setup employs a plume-rise parameterization originally designed for coarser grid simulations with resolutions around 100 km.

Furthermore, the model implementations are subject to considerable uncertainties. The first major source of uncertainty stems from the input variables provided by GFAS, which are derived from MODIS FRP measurements. These measurements are affected by interference from clouds and dense smoke plumes, suggesting a possible underestimation of fire intensity, leading to reduced aerosol, heat, and moisture emissions (Kaiser et al., 2012). Furthermore, coarse assumptions were made for the conversion of FRP to convective heat flux, acknowledging that this conversion remains highly uncertain (Val Martin et al., 2012). The same applies to the assumptions regarding the emitted moisture release.

In contrast to the typical diurnal cycle of atmospheric stability and fire intensity, which suggests pyroCb clouds form in the early to late afternoon, some of the most intense pyroCb activity during the ANY event was observed at night (Peterson et al., 2021). This discrepancy between the diurnal cycle of atmospheric stability, fire intensity and the nighttime pyroCb activity is not captured in the simulation and discussed in more detail by Muth et al. (2025).

Aerosol-radiation interactions from biomass burning can significantly influence plume dynamics, including plume rise, as shown by Ohneiser et al. (2023) and in our own set of simulations, and are highly sensitive to aerosol optical properties. We assume a OC/BC ratio in agreement with literature, however this ratio is strongly dependent on the vegetation type, combustion efficacy and moisture content (Janhäll et al., 2010). Further, the BC/OC ratio influences the balance between absorption and scattering, where higher BC content increases absorption and reduces SSA, while higher OC content enhances scattering. Additionally, in this study, aerosols are treated as internally mixed particles composed of an insoluble core and a soluble shell. Upon transitioning to this mixed state, particles are assigned fixed optical properties, though this approach carries substantial uncertainties. The aerosol size distribution plays a central role in determining radiative effects. Changes in size distribution due to aging processes such as condensation, coagulation, and chemical transformation can significantly alter AOD and radiative forcing (Seinfeld and Pandis, 2006), which is not accounted for. This limitation is particularly relevant in the context of pyroCb events, where recent findings indicate that black carbon particles injected into the stratosphere are heavily coated with organic material (Beeler et al., 2024). The ratio of soluble to insoluble mass affects scattering efficiency, that is again depending on the undergoing aging processes of the particle. Further, the inorganics-to-H<sub>2</sub>O ratio, depends on the aging stage and reflects the relative abundance of inorganic species to water and affects both refractive index and hygroscopic growth potential. While these ratios and the size distribution are held constant for computational efficiency, this simplification neglects the dynamic aging of aerosols, which alters both size and composition over time (Fierce et al., 2015). Furthermore, aerosol morphology








significantly impacts optical properties. Freshly emitted soot particles typically exhibit fractal, chain-like structures and are initially hydrophobic, yet this study assumes spherical morphology. According to Romshoo et al. (2022), such assumptions can lead to substantial overestimatio, up to a factor of five for SSA and up to a factor of three for the absorption coefficient. While the assumption of internally mixed aerosols with fixed optical properties simplifies model implementation, it limits the ability to capture the complexity of aerosol aging and the diversity of soot morphologies. Incorporating dynamic mixing state representations and morphology-aware optical models would improve the accuracy of aerosol–radiation interaction simulations, especially in regions influenced by fresh emissions and rapid chemical processing.

Additionally, the performed simulations do not account for aerosol-cloud interaction, which plays an important role in the formation of pyro-Cbs, as outlined in Fromm et al. (2022). While studies such as Andreae et al. (2004); Koren et al. (2005); Wang et al. (2009) report enhanced updrafts due to these interactions, Luderer et al. (2006) found that although aerosol loading significantly alters the microphysical structure of pyro-convective clouds, the influence of cloud condensation nuclei on the dynamic evolution of the pyroCb remains limited. More recent studies by Kablick III et al. (2018) indicate that the impact of fire-generated aerosols on the development of a specific pyroCb were negligible compared to the effects of fire-generated heat fluxes. Therefore, we assume the effect of aerosol-cloud interaction on the plume development is overall small in comparison to effects regarded in our simulation.

Further, the comparison with observations poses additional uncertainties. The NASA 3D wind retrieval itself is subject to uncertainties. The height retrievals depend on the relative viewing geometry of LEO-GEO. The retrieval process estimates the uncertainty of the retrieved parameters using a covariance matrix. This covariance matrix calculates the uncertainty statistics for, beyond other parameters, the retrieved height. The uncertainty derived from the covariance matrix serves as an effective guide to the quality of the retrievals (Carr et al., 2019). The shown retrievals are given with and error range of  $\pm$  200 to 300 m, which is consistent with the range reported by Carr et al. (2019).

Additionally, uncertainties persist in the comparison, as it is unclear at which specific mass mixing ratio or optical thickness the plume or cloud is detected by the satellite. Therefore, the comparisons with simulations are strongly dependent on the threshold chosen for the plume and cloud definition. Furthermore, the vertical resolution of the model, which increases with height, introduces a varying uncertainty. The model's vertical resolution in the altitudes between 9 and 15 km ranges from 200 to 250 meters. This is comparable to the error range of the observations. Further uncertainties are introduced by the analysis methods. As altitude increases, the grid cell heights become larger, and analysis may disproportionately represent higher altitudes, skewing the results. However, we use this approach because it provides a consistent method for identifying the highest plume altitudes across different scenarios.

#### 5 Conclusion

The Australian New Year's event serves as a test case for the analyses of the impact of intense fires in extreme meteorological variables. Our simulation captures the first phase of extreme pyro-convection during the first simulation day and the decline in fire activity over the following two days. We chose a resolution, at which it is not possible to resolve convection, so a






plume-rise model is used to parameterize the injection height. However, the resolution is fine enough that the impact of the fire on meteorological variables cannot be neglected. We demonstrate that for these intense fires, the implemented plume-rise model underestimates the injection height. The inclusion of fire-atmosphere interaction helps bridge this gap. The impact of fire-induced moisture is overall small for this test case. The fire-induced heat release shows that due to additional buoyancy, the injection height is increased, and cloud formation is possible without additional moisture release. Furthermore, it is shown that aerosol-radiative effects enhance the plume height downstream. The lofting is accompanied by additional cloud formation (semi-direct aerosol effect). Overall, the simulations accounting for aerosol-radiation interaction align better with the observations. The comparison to CALIPSO data indicates that all fire-atmosphere interactions need to be accounted for to reproduce a aerosol layer above 15 km, as observed two days after the extreme pyro-convection. Therefore, a combination of increased injection height and aerosol lofting is necessary. In conclusion, the strong impact of the fire on meteorology significantly influences the plume-rise model, enabling the emission of aerosols into the UTLS region for extreme fire events. Our results highlight the critical role of fire-induced heat release in accurately capturing the initial emission height of intense wildfires. We propose a simple parameterization based on GFAS data to account for this effect. Incorporating this mechanism not only improves the representation of plume rise but also enhances the simulation of aerosol-radiation interactions, which are amplified due to the increased emission height. Additionally, we show that the plume-rise model performs well for moderate fires, and the effects of fire-induced heat and moisture become negligible.

Code availability. The ICON and ART models are openly available at: https://icon-model.org/. The simulations in this study are based on a code version closely aligned with ICON release 2024.10 (https://doi.org/10.35089/WDCC/IconRelease2024.10). Certain model components used in this work, which are not fully open-source, can be provided upon reasonable request to the corresponding author. Access to the NASA 3D Wind Algorithm was granted by Dr. James Carr (jcarr@carrastro.com) and remains subject to his approval. Analysis and visualization scripts were adapted from: https://github.com/alihoshy/art\_pytools.

Data availability. The ICON-ART simulation output generated in this study will be made available via Radar4KIT with a DOI following the peer-review process. Himawari-8 satellite data are publicly accessible through Amazon Web Services (AWS): https://registry.opendata.aws/noaa-himawari. MODIS datasets were obtained from: https://ladsweb.modaps.eosdis.nasa.gov/search/order/1/MODIS. CALIPSO data are publicly available and were downloaded from: https://asdc.larc.nasa.gov/project/CALIPSO. Access to the standardised database of quality assured air pollution monitor data from Australian state and territory governments is available to researchers on request to car.data@sydney.edu.au

Figure A1. LWP for the a) REF, difference in LWP for b) ARI-REF, and c) ALL-REF on December 31 at 14:30 AEDT.

Figure A2. IWP for the a) REF, difference in IWP for b) ARI-REF, and c) ALL-REF on December 31 at 14:30 AEDT.

## Appendix A

Author contributions. L.M.: Conceptualization, Methodology, Software, Validation, Visualization, Writing – original draft, Writing – review 630 & editing. C.H.: Supervision, Writing – review & editing. H.V.: Software, Validation, Writing – review & editing. B.V.: Conceptualization, Supervision, Writing – review & editing. G.A.H.: Conceptualization, Methodology, Software, Supervision, Writing – review & editing.

Competing interests. The authors declare that they have no competing interests.


Figure A3. Plume top height for the a) ARI, b) ALL, c) REF, d) MOIST, e) HEAT on December 31 at 14:30 AEDT.

Acknowledgements. This study contains modified Copernicus Atmosphere Monitoring Service information [2023]. This research was supported by resources provided by the Deutsches Klimarechenzentrum (DKRZ) under project ID bb1070. The study also received funding through the project *PermaStrom* (grant no. 03EI4010B), part of the seventh energy research program of the German Federal Ministry for Economic Affairs and Climate Action (BMWK). L.M. acknowledges support from the Graduate School for Climate and Environment (GRACE). We thank the German Weather Service (DWD) for providing meteorological analysis products. We also acknowledge the use of the NASA 3D Wind Algorithm developed by Dr. James Carr and colleagues at NASA Goddard Space Flight Center, particularly Dr. Mariel Friberg and Dr. Dong Wu, whose contributions were instrumental in the analysis of plume and cloud top heights. During the preparation of this manuscript, the authors used Microsoft Copilot to assist with writing and formulation. All content was subsequently reviewed and edited by the authors, who take full responsibility for the final version of the manuscript.

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
