# Peer review of "Integrating Fire-Induced Meteorological Changes into Plume Rise Modeling for Extreme Wildfire Simulations"

_EGUsphere, 2025_

## Referee Comment (RC1)

In this study, Muth et al. incorporate fire-induced meteorological changes into plumerise modeling and examine the effects of fire-atmosphere coupling within the host model on plume development. The results demonstrate that fire-released heat plays a primary role in increasing plume height, mainly through enhanced buoyancy and cloud formation. In contrast, moisture released by fires has a negligible influence on plume dynamics. Beyond sensible and latent heat fluxes, aerosol-radiation interactions exert contrasting effects on plume injection height: initially, they reduce injection height by stabilizing the atmosphere, but subsequently increase it via a lofting effect. Overall, this work represents a significant advancement in the modeling of fire-atmosphere interactions. The findings align with previous studies in the literature. I think that the manuscript is suitable for publication in ACP once the following points are adequately addressed.

It is unclear how the calculated fire-induced sensible heat flux and moisture release are incorporated into the fire plume rise model and how they affect the grid-scale meteorological fields of the host model. For instance, is the fire-induced sensible heat flux added directly to the surface heat flux in the host model, or is it converted into temperature perturbations relative to the environmental temperature in the plume? Similarly, is the fire-induced moisture release included in the plume's water vapor content? If both terms are introduced into the governing equations of the plume model (it may be helpful to present these equations), how do they subsequently influence the air temperature and moisture fields in the host model?

Additionally, how is aerosol–radiation feedback incorporated into the fire plume rise model? Is this process represented in the governing equations for plume temperature and vertical velocity? Furthermore, how is this feedback subsequently transmitted to the host model—specifically through the direct radiative effects of fire aerosols? At present, many key details appear to be omitted. Thus, the authors should provide a clear and explicit description of these representations.

In the model configuration, the limited-area model simulations are conducted with a grid spacing of 6.6 km. Are the calculated fire-induced sensible flux and latent heat flux (i.e., the moisture release) upscale to be relative to the grid area or still relative to the burned area (i.e., sensible/latent heat per unit grid area or that per unit burned area)? As the authors noted, the grid spacing is still too coarse to explicitly resolve convection and the associated plume-rise processes, which, to some extent, will smooth out the fire heat effects. In this context, a recent study by Ma et al. (2025) innovatively addressed this issue by incorporating fire heat at the subgrid scale within the convection scheme of a global climate model. Regarding this point, I recommend that the authors include a thorough discussion of this in the final section of the manuscript.

Regarding Eq. (1) for estimating fire size within a grid cell: why did the authors not directly use observed burned area products? Equation (1) assumes a linear relationship between FRP and burned area, which may not hold in all cases. For instance, although

forest fires typically burn smaller areas than grass fires, they can exhibit higher FRP due to denser fuel loads (Zheng et al., 2021). Moreover, the triggering of convection is influenced by FRP density (i.e., FRP per unit burned area), rather than by total FRP (i.e., total fire heat or fire heat per unit grid area). The current approach may therefore underestimate the capacity of forest fires to initiate convection, given their high sensible heat intensity (i.e., sensible heat per unit burned area) (Ma et al., 2025).

It is unclear whether the parameters of the diurnal cycle function are applicable to Australian wildfires, or if they vary across different climate zones and fire regimes. Further clarification on this point would strengthen the manuscript.

The parameters in Eq. (7) should ideally be dependent on the fire regime. It would be more appropriate to integrate moisture release from different fire types using a lookup table, following the approach of Ma et al. (2025). Alternatively, the authors should provide a comprehensive discussion of the uncertainties associated with the current simplification.

I suggest reorganizing the manuscript to improve the logical flow. Presenting Section 3.2 first would be more effective, as it demonstrates that the ALL simulation agrees well with observations. Establishing the model's credibility upfront would provide a stronger foundation for presenting the sensitivity experiments in Section 3.1 for the mechanism analysis.

In the final section, I recommend that the authors include a discussion of the uncertainties in this study, including some of the issues raised earlier in this review.

Regarding the experimental design: the REF simulation was conducted as a global run at 13 km resolution, which differs from the 6.6 km grid spacing used in the limited-area simulations. This discrepancy in resolution may introduce additional uncertainty. To better isolate the impact of fires, it would be valuable to also include a limited-area simulation without fire influences for a more consistent comparison.

Figure 1: Consider adding a panel to schematically illustrate how the revised plumerise model is coupled in ICON-ART to account for fire-induced meteorological feedbacks to the host model.

Figures 6 and 7: Why are there limited clouds over the biomass burning regions, despite the expected fire-induced convection? Also, suggest adding wind vectors to denote the downstream direction.

Figures 8 and 9: It seems that all the simulations show substantial discrepancies compared to observations.

**References:**

1. Ma, Q., Wei, L., Wang, Y., Zhang, G. J., Zhou, X., & Wang, B. (2025). Fire heat

- affects the impacts of wildfires on air pollution in the United States. *Science*, 389(6765), 1137-1142.
- 2. B. Zheng et al., Increasing forest fire emissions despite the decline in global burned area. *Sci. Adv.* 7, eabh2646 (2021).

---

## Referee Comment (RC2)

The study titled "Integrating Fire-Induced Meteorological Changes into Plume Rise Modeling for Extreme Wildfire Simulations" uses the Freitas et al. model to simulate plume rises during the Australian 2019/2020 wildfire season. The authors included fire-induced moisture releases and other fire-atmosphere feedbacks. Smoke simulations were improved when the ICON/Freitas model included radiative aerosol effects, and heat/moisture releases.

**General comments:**

Overall, the results of this study are interesting and are highly relevant for global/regional aerosol transport models and I believe this work will eventually be suitable for publication. The impacts of aerosol radiative effects on smoke lofting height is an especially neat result and separates this study from the existing literature. With that said, the methodology section of this manuscript could use significant improvements, and it's unclear how the Freitas model was incorporated within ICON, and how moisture fluxes and aerosols radiative fluxes are implemented, and the order in which these processes are calculated. A lot of these details need to be clarified in the methodology before this manuscript can be considered ready for publication.

The introduction felt a bit "long-winded," and I wonder if there's text that could be omitted. The authors may want to take a closer look at the content between lines 75–115 to see whether some material could be trimmed or written in a more concise and consolidated way.

For the methodology section, it was unclear how moisture fluxes and aerosol radiative effects were implemented within the Freitas et al. plume-rise model. For example, was an aerosol-radiative-effect subroutine or parameterization added directly into the Freitas model? Or is smoke first lofted by the Freitas model and then inserted into the 3D fields of the ICON model, where aerosol radiative effects within ICON further loft the smoke aerosols through "radiative lofting"? Along these lines, how were moisture fluxes incorporated into the Freitas et al. model? Were they used to modify the moisture content in the first vertical level of the Freitas model? This seems to be implied by the statement that "the moisture is added to the specific humidity tracer," but it's unclear whether this refers to ICON or Freitas. More details on how moisture fluxes were coupled with the Freitas model would be helpful. I also wonder whether Figure 1 could be used more effectively to describe the workflow and the order in which aerosol radiative effects and moisture fluxes are computed in the ICON/Freitas plume-rise framework. Figure 1 may also need to be reworked—it's a bit vague in its current form.

The abstract and introduction discuss how fire—atmosphere interactions were implemented within the ICON/Freitas plume-rise modeling framework. I believe the manuscript uses the term "fire-induced meteorological effects." However, I did not see any explanation of how fire—atmosphere interactions were included in this framework. When I think about fire—atmosphere interactions, I think of models like WRF-FIRE or WRF-SFIRE, where the fire generates a plume, and through mass continuity the near-surface winds accelerate, causing the fire to spread faster and release more energy, which then further strengthens the plume-rise updraft. I suspect many fire modelers interpret "fire—atmosphere interactions" in this way. If these types of feedbacks are not included in the ICON/Freitas framework, the authors may want to revise the abstract to more clearly define what is meant by fire—atmosphere or meteorological interactions.

The structure of the discussion section could also be improved. Some of the descriptions seem to bounce around a bit, and it may be worthwhile for the authors to give this section another pass to make it read more coherently.

**Specific comments:**

Line 20: While I suspect this # is not too far off, it's worth noting that these analyses were based on MISR data, which corresponds to satellite overpasses that occur during the late morning/early afternoon, when plumes are not fully developed. Granted, the PBL was also probably shallower during these times so the % above the PBL may not change a ton, but maybe just note that these overpasses occur in the late morning/early afternoon.

Line 40: Another issue, especially for the empirical-based Sofiev scheme is that it's trained off MISR data and does a poor job with extreme plume rise values, e.g., the tail of the distribution. It does better with smaller plumes, but it's usually not the small plumes we care about. The larger fires that emit orders of magnitude more smoke are generally the ones that generate the plumes that loft smoke further into the free troposphere.

Line 60: I would argue the Freitas model is computationally efficient. As a standalone model, it runs in less than 0.1 seconds on a single CPU core. ML/AI approaches could be faster, but in terms of computational time relative to the host 3D model my guess is this would not really have much of an impact on the model run time.

Line 160: replace "this limitation" with "cloud blocking", or something along these lines to make this clearer of what "this" is referring to.

Line 175: Slightly confused here, it was noted that geostationary satellites can observe the diurnal cycle, but it looks like an alternative cyclic function was used? It was unclear if this cyclic function was related to geostationary satellites observations or not.

Line 187: I am sure this OK, but why not use a mass flux approach to directly compute the smoke detrainment (see Wilmot et al. 2022)? This gets away from the parabolic emission profile assumption, which is a little "hand wavy", and uses an approach based on plume physics.

Wilmot, K., D. V. Mallia, A. G. Haller, and J. C. Lin, 2022: Wildfire plumes in the Western US are reaching greater heights and injecting more aerosols aloft under a changing climate. Scientific Reports, 12, 12400.

Line 209: Where did the 3.4 GFAS correction factor come from? It looks like this is based on Kaiser et al., so are we just assuming that since the smoke emissions are off by a factor of 3.4, this difference is related to a 3.4 difference in fuel consumption, which would be proportional to heat fluxes? Is it possible that this 3.4 might be the result of uncertainty with fire emission factors? These emission factors, especially for PM2.5 can be quite large (see Urbanski 2014). It might be good to include some of the limitations of this assumption at the very least.

*Urbanski, Shawn. 2014. Wildland fire emissions, carbon, and climate: Emission factors. Forest Ecology and Management. 317: 51-60.*

Line 209: In equation 6, what is the 5.5? Did you mean 0.55?

Line 310: See the general comment, but it is unclear how moisture fluxes and aerosol—radiation interactions were incorporated in ICON/Freitas, along with the order of these calculations. It is also unclear what is meant by a fire-induced heat flux? If I remember correctly, the Freitas et al. plume rise model will not run if the Heat flux is equal to 0. If the "fire induced heat flux" is 0, is the plume rise model just being forced by terrestrial radiative fluxes?

Line 345: Are heights reported in ASL or AGL? Might be good to add this when reporting heights here, i.e., 3.5 kmAGL.

Line 343: How does the fire destabilize the atmosphere in ICON? I do know that the Freitas model runs for X time steps to develop to build up the plume (I think this takes anywhere from 30-55 model minutes)? Is ICON reading output from each time step from the Freitas model and using that to modify the 3D weather/aerosol fields within ICON? Or are the emissions only released vertically once the plume has reached steady state? Along these lines, is the Freitas model one-way or two-way coupled with ICON, both in terms of meteorology/aerosols?

Line 424: The plume is rising— is this after the smoke has been vertically lofted by the Freitas model, i.e., has the smoke been carried upward by the Freitas model, vertically distributed within the ICON vertical column, and then lofted up further due the aerosol radiative heating that is resolved within ICON?

Line 429-434: Near-surface aerosol layers promote stability by warming low/mid-level layers, which makes sense. But I am not entirely following the upper-level description? If the smoke is lofted further up, would there still be warming aloft and therefore a stable layer, which is just higher up in the atmosphere?

Line 455: Okay, I think this loosely answers my earlier question about the order of operations for the aerosol radiative feedbacks. However, this really needs to be explained more clearly in the methods section. I strongly recommend redesigning Figure 1 so it clarifies the sequence of steps: the heat-flux/emission calculation -> the Freitas model simulation -> injection of smoke into ICON -> the physics that ICON resolves once the smoke fields have been added. It seems that the same workflow applies to moisture as well. While this order makes sense for aerosol radiative feedbacks, I wonder whether releasing moisture after the plume-rise calculation is a potential limitation? Moisture fluxes could be relevant for the plume-rise calculation itself. For example, if the lower boundary conditions in the Freitas model contained more moisture, the rising plume would reach the lifting-condensation level sooner, allowing latent-heat release to occur at a lower altitude. That additional buoyancy could enable the plume to rise higher. Coupled fire—atmosphere simulations have explored this effect, and although some preliminary results suggest that moisture fluxes may not matter much for plume development, I still wonder if adding moisture only after the plume-rise step is an oversimplification. This is, of course, assuming I'm interpreting the order of operations in the Freitas/ICON framework correctly.

Line 470: Does ICON resolve SOA formation? Might be good to note this somewhere (maybe the methods, unless I missed this). Seems like there is no SOA based on the statement in line 563.

Line 535: This seems to be a common issue with plume-rise models and has been documented for the Freitas model as well. However, I'm not sure we can attribute the underestimation solely to the plume-rise model itself. The inputs to these models, namely heat flux and active fire area, are highly uncertain, and the model is very sensitive to heat-flux density. I'm not confident that we have a solid understanding of the actual heat-flux density for wildfires outside of a few field campaigns, since current satellite fire detection data is likely too coarse to definitively determine whether plume-rise models truly have a systematic underestimation problem. Maybe we are just systematically underestimating the heat flux density? This point goes beyond the scope of the paper, and satellite-derived inputs remain one of the best available option despite their limitations. Still, it may be worth noting this somewhere in the discussion or conclusion section. Along these lines, while the addition of moisture and aerosol physics within the Freitas/ICON framework appears to improve the results, a major caveat is that these enhancements may be compensating for systematic underestimation of how plume-rise inputs (heat flux and fire area) are computed. In any case, this isn't something I expect the authors to resolve—it's just a thought. I believe this idea is somewhat alluded to around line 546.

---

## Author Comment (AC1)

**Response to reviews**

We would like to express our gratitude to the referees for their thorough review and valuable feedback on our manuscript. We have carefully considered all the comments and suggestions provided. Below, we present a detailed point-by-point response to each comment, specifying the changes made in the revised manuscript. We believe these revisions have significantly improved the quality and clarity of our work. In the following the reviewers comments are bold, for better overview.

**RC1**: https://doi.org/10.5194/egusphere-2025-4853-RC1

**In this study, Muth et al. incorporate fire-induced meteorological changes into plume-rise modeling and examine the effects of fire–atmosphere coupling within the host model on plume development. The results demonstrate that fire-released heat plays a primary role in increasing plume height, mainly through enhanced buoyancy and cloud formation. In contrast, moisture released by fires has a negligible influence on plume dynamics. Beyond sensible and latent heat fluxes, aerosol–radiation interactions exert contrasting effects on plume injection height: initially, they reduce injection height by stabilizing the atmosphere, but subsequently increase it via a lofting effect. Overall, this work represents a significant advancement in the modeling of fire–atmosphere interactions. The findings align with previous studies in the literature. I think that the manuscript is suitable for publication in ACP once the following points are adequately addressed.**
**It is unclear how the calculated fire-induced sensible heat flux and moisture release are incorporated into the fire plume rise model and how they affect the grid-scale meteorological fields of the host model.**

R: We acknowledge that the previous version of the methodology section lacked sufficient detail, which may have led to misunderstandings. To clarify: The implementation of heat release occurs within the ICON-ART model. This affects atmospheric variables that are subsequently read by the plume rise model, leading to changes in calculated plume heights compared to simulations without heat release in the host model. No explicit modifications were made to the Freitas model itself. Similarly, additional moisture is released in ICON-ART (again, without changes to the Freitas model). This alters the meteorological input provided to the Freitas model, which in turn influences plume height calculations. Aerosol-radiation interactions are also accounted for only in ICON-ART. The resulting radiative changes impact meteorological variables read by the Freitas model, thereby indirectly affecting the calculated emission heights.
We have revised the methodology section to include these clarifications and added statements describing the workflow.

l198: "In this study, we use the term "fire-induced meteorological changes" to describe modifications, which are induced by fire-related heat and moisture release and/or aerosol-radiation in the ICON-ART model. Thereby, the plume-rise model remains unchanged; it is only influenced by the modified atmospheric state provided by ICON-ART. Figure 3 illustrates how ICON-ART supplies atmospheric input to the plume-rise model. Within the emission routine, the plume-rise model calculates the emission height based on the atmospheric conditions from ICON-ART and the vegetation type of the corresponding grid cell. The plume-rise model internally accounts for latent and sensible heat from the fire as well as cloud formation, but these meteorological changes are not fed back to the ICON host model. Consequently, without the modifications applied in this work, the host model remains unaware, except for the emitted aerosols. In this study, we analyze how incorporating fire-induced heat and moisture release, together with aerosol–radiation interactions, affects atmospheric conditions in ICON-ART and how these changes ultimately influence the calculated plume height, without any tuning of the Freitas model itself. "

l264: "The calculation of aerosol radiative effects takes place in in the ICON-ART model. Aerosol-radiative effects impact the atmospheric stability close to the fire source, which is used by the plume rise model to calculate the emission height, as well as downstream of the plume. Other than the modified atmospheric profiles, the Freitas model does not account for aerosol-radiation interaction."

Further we added Section 3.1, which shows how ICON-ART variables are impacted by the implementations.

**For instance, is the fire-induced sensible heat flux added directly to the surface heat flux in the host model, or is it converted into temperature perturbations relative to the environmental temperature in the plume?**

We clarified this:
l214: "The resulting heat release is implemented as a sensible heat flux from the surface to the atmosphere within the land/surface scheme. ICON-ART then takes this additional heat flux into account when calculating the prognostic variables. "

**Similarly, is the fire-induced moisture release included in the plume's water vapor content?**

R: Yes, the water water vapor content within the plume (host model) changes.

l230: "The moisture is added to the specific humidity tracer in the ICON-ART model. The moisture emission follows the same injection height and profile as the particles, according to the plume-rise model. Therefore, the moisture-release emission rate $E_{qvfire}$ can obtained by inserting $qv_{fire}$ as $M$ in Equation 5. As, this additional moisture is added to the existing qv (specific humidity) tracer of ICON and thus, contributes to all physical and dynamical processes. The changes in the atmospheric state by either the heat or moisture release thereby change the atmospheric profiles that are inputs for the plume rise model, and therefore impact the calculated injection height."

**If both terms are introduced into the governing equations of the plume model (it may be helpful to present these equations), how do they subsequently influence the air temperature and moisture fields in the host model?**

R: Ultimately, the changes are introduced into the governing equations of ICON. The heat release is added within the land-surface scheme as a sensible heat flux from the surface to the atmosphere, within ICON-ART. And the moisture is added directly to the qv tracer. While we considered including the full set of governing equations, we believe this would shift the focus too much toward technical implementation rather than clarifying the workflow. However, we refer to Zängl et al. (2015), who describes the governing ICON equations. Figure R1 provides a clearer conceptual overview without delving into the full equation set.

**Additionally, how is aerosol–radiation feedback incorporated into the fire plume rise model? Is this process represented in the governing equations for plume temperature and vertical velocity? Furthermore, how is this feedback subsequently transmitted to the host model—specifically through the direct radiative effects of fire aerosols? The radiative feedback is only accounted for in the host model At present, many key details appear to be omitted. Thus, the authors should provide a clear and explicit description of these representations.**

R:Aerosol–radiation feedback is not incorporated into the Freitas plume-rise model. Instead, it is fully represented within the ICON-ART host model through its radiation scheme. For aerosols, optical properties are provided directly to the radiation scheme (ecRad) via lookup tables. Using these properties and the local aerosol mass concentration at each grid point and level, ecRad calculates the radiative transfer parameters—optical depth, single-scattering albedo, and asymmetry parameter. These calculations are performed in the ART radiation routine, which is called every time ecRad is invoked. ecRad then computes reflection, transmission, and internal radiation sources online for each grid box and model level, resulting in upward and downward radiative fluxes. In addition, it calculates radiative heating and cooling rates, which feed back into the dynamics and physics of ICON-ART. This process is schematically shown in Figure R1. We have added this explicit description to the manuscript to clarify how aerosol–radiation interactions are represented and how they influence the host model.

l118: "For aerosols, optical properties are supplied directly to the radiation scheme via lookup tables. The radiative transfer parameters (optical depth, single scattering albedo and asymmetry parameter) are obtained by using the aerosol optical properties and the local aerosol mass concentration. The cal-

[Figure]

R 1: Coupling of the dynamical core, the NWP physics, and ART routines package in ICON-ART, adapted from Rieger et al. (2015) and extended by modifications in this work.

culation of the radiative transfer parameters is performed in the radiation routine and passed to ecRad. ecRad calculates reflection, transmission, and internal radiation sources online for each grid box and model level, resulting in the upward and downward radiative fluxes. In addition, it calculates radiative heating and cooling, which feeds back into the dynamics and physics of ICON."

**In the model configuration, the limited-area model simulations are conducted with a grid spacing of 6.6 km. Are the calculated fire-induced sensible flux and latent heat flux (i.e., the moisture release) upscale to be relative to the grid area or still relative to the burned area (i.e., sensible/latent heat per unit grid area or that per unit burned area)?**

R: The upscaling is according to grid unit area. We clarify this here:
l319: "As mentioned, GFAS has a resolution of 0.1° and is provided on a regular longitude-latitude grid, and therefore needs to be remapped to the ICON triangular grid. This is done using the Radial Basis Function Interpolation."

**As the authors noted, the grid spacing is still too coarse to explicitly resolve convection and the associated plume-rise processes, which, to some extent, will smooth out the fire heat effects. In this context, a recent study by Ma et al. (2025) innovatively addressed this issue by incorporating fire heat at the subgrid scale within the convection scheme of a global climate model. Regarding this point, I recommend that the authors include a thorough discussion of this in the final section of the manuscript.**

R: We have added this to the discussion:
l645: "Fire-induced heat and moisture release were distributed uniformly across each grid cell, rather than applied at subgrid scale, as proposed by Ma et al. (2025). In their approach, fire heat was explicitly represented within the superparameterized Community Earth System Model (SP-CESM), which embeds 2-dimensional cloud-resolving models (CRMs) at 4 km resolution inside each GCM grid cell. Fire heat fluxes were applied only to a subset of CRM columns proportional to the burned area, creating horizontal heterogeneity and localized buoyancy maxima that drive secondary circulations and more realistic convection. In our simulation the grid spacing of 6.6 km is comparable to the 4 km resolution of the CRMs. Further, the GFAS input resolution (0.1°, ≈11 km) allows no further subdivision. Nevertheless, neglecting localized buoyancy maxima may dilute effective heat-flux density and can reduce injection heights. However, injection height is parameterized within the plume rise model, and the implementation of heat

and moisture release in ICON-ART allows for an averaged fire effect on meteorological variables in the host model, ensuring that large-scale atmospheric responses to fire heat are still represented."

**Regarding Eq. (1) for estimating fire size within a grid cell: why did the authors not directly use observed burned area products? Equation (1) assumes a linear relationship between FRP and burned area, which may not hold in all cases. For instance, although forest fires typically burn smaller areas than grass fires, they can exhibit higher FRP due to denser fuel loads (Zheng et al., 2021).**

R: The chosen approach is a published method by Val Martin et al. (2012), which is also used by Ke et al. (2021), for scaling burned area based on FRP, which enables the use of GFAS data already provided for other components of the implementation. We acknowledge that this method introduces uncertainties, as the assumed linear relationship between FRP and burned area may not hold universally. However, since the vegetation burned during the ANY event consisted almost exclusively of eucalyptus forest and the scaling is applied locally, a linear relationship can reasonably be assumed in this case.
Since our model setup relies on GFAS for emissions, we aimed to maintain consistency across all inputs. Applying observed burned-area products would indeed be an interesting test case for future work, and we have noted this in the discussion.

l621: "Second, Equation 1 assumes a linear relationship between FRP and burned area, which may not hold in all cases. For example, although forest fires typically burn smaller areas than grass fires, they can exhibit higher FRP due to denser fuel loads (Zheng et al., 2021)."

**Moreover, the triggering of convection is influenced by FRP density (i.e., FRP per unit burned area), rather than by total FRP (i.e., total fire heat or fire heat per unit grid area). The current approach may therefore underestimate the capacity of forest fires to initiate convection, given their high sensible heat intensity (i.e., sensible heat per unit burned area) (Ma et al., 2025).**

R: This is correct. However, for the extreme event analyzed in this study—where the fire size is much larger than the grid spacing and the overall fire intensity is very high—the current approach captures convection reasonably well. For coarser resolutions, sub-grid parameterization would indeed be beneficial, and we acknowledge this as an important consideration for future work.

**It is unclear whether the parameters of the diurnal cycle function are applicable to Australian wildfires, or if they vary across different climate zones and fire regimes. Further clarification on this point would strengthen the manuscript.**

R: We have revised the methodology section to include a discussion on the applicability of the diurnal cycle parameters. In this context, we refer to the work of Muth et al. (2025), which provides a detailed discussion of the topic. Additionally, we have included a figure in Appendix A that illustrates the discrepancies and supports the explanation provided.

l169: "The diurnal cycle of fires varies considerably across regions and depends on both fuel type and meteorological conditions. For instance, extreme fires often show pronounced peak activity, extended duration throughout the day, and persistent nighttime burning. Despite these variations, a majority of vegetation fires follow a characteristic diurnal cycle that can be approximated by a Gaussian distribution, typically peaking in the early afternoon (Vermote et al., 2009). A generalized diurnal cycle function has been proposed by Kaiser et al. (2009) and Andela et al. (2015) and was applied by Walter et al. (2016) in COSMO-ART; this approach is also implemented in ICON-ART. The diurnal cycle function, denoted as $d(t_1)$, is applied to both fire intensity and fire size within the plume-rise model."

l623: "Next, applying a typical diurnal cycle function introduces uncertainties that vary across different climate zones and fire regimes. Weighting the diurnal cycle according to vegetation type is an important first step; however, impacts like combustion phases, for example, are neglected. Additionally, in contrast to the typical diurnal pattern of atmospheric stability and fire intensity—where pyroCb clouds generally form in the early to late afternoon—some of the most intense pyroCb activity during the ANY event occurred at night (Peterson et al., 2021). This discrepancy between the expected diurnal

[Figure]

R 2: The averaged FRP assumed in the simulations (black line), and the standard deviation in gray without enhancement factor and FRP with 3.4 enhancement factor in yellow, with the standard deviation in dark yellow. Red dots represent the mean MODIS active FRP (NASA and University of Maryland, 2025) for each hour with measurements during the first simulation day (December 30, 2029), along with the standard deviation

cycle of atmospheric stability, fire intensity, and observed nighttime pyroCb activity is not captured in the simulation and is discussed in more detail in Muth et al. (2025), and can also be seen in Appendix A."

**The parameters in Eq. (7) should ideally be dependent on the fire regime. It would be more appropriate to integrate moisture release from different fire types using a lookup table, following the approach of Ma et al. (2025). Alternatively, the authors should provide a comprehensive discussion of the uncertainties associated with the current simplification.**

R: This is correct. For this case study, the assumption is based on published values for southeastern Australia and is therefore explicitly tailored to this event rather than being generally applicable. We acknowledge that this simplification introduces uncertainty and have added a discussion to highlight this limitation. Incorporating moisture release from different fire types using a lookup table, as suggested by Ma et al. (2025), would be an important improvement for future work.
We clarified this in the discussion:
l633: "Moisture release estimates introduce additional uncertainty, as weighting factors in Equation ?? are based on fuel-moisture characteristics from Nolan et al. (2016) and Deb et al. (2020) for southeast Australia, they may not be generally applicable. "

**I suggest reorganizing the manuscript to improve the logical flow. Presenting Section 3.2 first would be more effective, as it demonstrates that the ALL simulation agrees well with observations. Establishing the model's credibility upfront would provide a stronger foundation for presenting the sensitivity experiments in Section 3.1 for the mechanism analysis.**

R: Thank you for this suggestion. We have reorganized the manuscript accordingly. Specifically, we now present an analysis of the isolated effects of heat, moisture, and aerosol–radiation interactions in Section 3.1 to establish the overall functionality of the implementation. This is followed by a comparison with observations and an analysis of the underlying mechanisms. Please refer to Section 3 for details.

**In the final section, I recommend that the authors include a discussion of the uncertainties in this study, including some of the issues raised earlier in this review.**

R: We have thoroughly revised and restructured the entire discussion section, addressing the issues you raised. We believe these changes adequately resolve the uncertainties highlighted. Please refer to Section 5 for details.

**Regarding the experimental design: the REF simulation was conducted as a global run at 13 km resolution, which differs from the 6.6 km grid spacing used in the limited- area simulations. This discrepancy in resolution may introduce additional uncertainty. To better isolate the impact of fires, it would be valuable to also include a limited-area simulation**

[Figure]

R 3: Schematic illustration of the coupling between the plume-rise model and ICON-ART. The diagram shows the sequence of processes: fire-induced sensible heat release, emitted moisture, and aerosol–radiation interaction within ICON-ART, which modify the atmospheric state. These updated conditions are then passed to the Freitas plume-rise model for plume height calculation. Finally, aerosol and moisture emissions are applied, and the cycle repeats.

**without fire influences for a more consistent comparison.**

R: We would like to emphasize that the global 13 km simulation not one of the experiments analyzed for this study. It is required to generate the meteorological boundary conditions for the limited-area model experiment simulations, including the REF experiment. Further, we would like to clarify that the REF simulation already represents a case without fire-induced meteorological changes. In this setup, the plume-rise model emits aerosols as passive tracers only; there are no aerosol–radiation or cloud interactions, and therefore the atmospheric state remains unaffected by the fire. This ensures that the REF simulation serves as a suitable baseline for comparison. We clarified this in:

l302: "Prior to the experimental limited-area mode simulations, a global simulation with a grid spacing of 13 km is performed to obtain the input data for the boundary conditions. This global simulation is initialized using the German Weather Service (DWD) analysis product and does not account for fire impacts on the meteorological variables. "

l332: "We conducted five experiments (Table 1: a reference run (REF), in which fire aerosol emissions are enabled and the plume-rise model calculates emission heights based on ICON-ART atmospheric conditions without accounting for any fire-induced impacts. In this configuration, aerosol–radiation interactions are not considered, and aerosols are transported as passive tracers. "

**Figure 1: Consider adding a panel to schematically illustrate how the revised plume-rise model is coupled in ICON-ART to account for fire-induced meteorological feedbacks to the host model.**

R: We have revised Figure 1 to better illustrate the workflow. For clarification, the plume-rise model itself was not modified; rather, the changes were implemented in ICON-ART, which altered the input provided to the plume-rise model.

**Figures 6 and 7: Why are there limited clouds over the biomass burning regions, despite the expected fire-induced convection? Also, suggest adding wind vectors to denote the downstream direction.**

R: Figures 6 and 7 (revised manuscript) represent conditions on December 31, when the maximum FRP was only 47 W m$^{-2}$. Our additional analysis (see Figure 5-6, revised manuscript) shows that the impact of heat and moisture release is negligible for fires smaller than approximately 25 W m$^{-2}$. Since only a few grid cells exceeded this threshold, significant pyroconvective activity was not expected on the second simulation day, which is consistent with reports of the ANY event. What we observe instead is plume generation and downstream transport of the previous day.

Regarding wind vectors: while we agree that indicating downstream transport would be useful, Figures 6 and 7 display column-integrated values and top heights, representing plumes across multiple vertical levels. Showing a single wind direction would therefore be misleading, and displaying multiple streamlines would overload the figure.

**Figures 8 and 9: It seems that all the simulations show substantial discrepancies compared to observations.**

R: We acknowledge that there are considerable discrepancies between the simulations and the observations. However, based on our experience and in-depth discussions with satellite experts, we are confident that these differences fall within the expected range when comparing model outputs with satellite data. Furthermore, we refer to our added discussion of the limitations and the underlying uncertainties, including missing aerosol–cloud interactions, omission of other aerosol species, the use of total rather than attenuated backscatter, and the absence of cloud backscatter. Despite these limitations, we observe a clear improvement in Figure 9 (revised manuscript) and a small effects in Figure 10 (revised manuscript), which supports the main point we intended to demonstrate.

l430: "The near-surface signal is primarily classified as marine aerosol with localized cloud contributions. Our simulations only account for wildfire aerosols; therefore, the absence of the maritime aerosol layer is expected. Similarly, the lack of clouds can be explained by the omission of aerosol–cloud interactions in the model configuration."

RC2: https://doi.org/10.5194/egusphere-2025-4853-RC2

**The study titled "Integrating Fire-Induced Meteorological Changes into Plume Rise Modeling for Extreme Wildfire Simulations" uses the Freitas et al. model to simulate plume rises during the Australian 2019/2020 wildfire season. The authors included fire-induced moisture releases and other fire-atmosphere feedbacks. Smoke simulations were improved when the ICON/Freitas model included radiative aerosol effects, and heat/moisture releases.**

**General comments: Overall, the results of this study are interesting and are highly relevant for global/regional aerosol transport models and I believe this work will eventually be suitable for publication. The impacts of aerosol radiative effects on smoke lofting height is an especially neat result and separates this study from the existing literature. With that said, the methodology section of this manuscript could use significant improvements, and it's unclear how the Freitas model was incorporated within ICON, and how moisture fluxes and aerosols radiative fluxes are implemented, and the order in which these processes are calculated. A lot of these details need to be clarified in the methodology before this manuscript can be considered ready for publication.**

R: To clarify: The implementation of heat release occurs within the ICON-ART model. This affects atmospheric variables that are subsequently read by the plume rise model, leading to changes in calculated plume heights compared to simulations without heat release in the host model. No explicit modifications were made to the Freitas model itself. Similarly, additional moisture is released in ICON-ART (again, without changes to the Freitas model). This alters the meteorological input provided to the Freitas model, which in turn influences plume height calculations. Aerosol-radiation interactions are also accounted for only in ICON-ART. The resulting radiative changes impact meteorological variables read by the Freitas model, thereby indirectly affecting the calculated emission heights.
We have revised the methodology section to include these clarifications and added statements describing the workflow.

l198: "In this study, we use the term "fire-induced meteorological changes" to describe modifications, which are induced by fire-related heat and moisture release and/or aerosol-radiation in the ICON-ART model. Thereby, the plume-rise model remains unchanged; it is only influenced by the modified atmospheric state provided by ICON-ART. Figure 3 illustrates how ICON-ART supplies atmospheric input to the plume-rise model. Within the emission routine, the plume-rise model calculates the emission height based on the atmospheric conditions from ICON-ART and the vegetation type of the corresponding grid cell. The plume-rise model internally accounts for latent and sensible heat from the fire as well as cloud formation, but these meteorological changes are not fed back to the ICON host model. Consequently, without the modifications applied in this work, the host model remains unaware, except for the emitted aerosols. In this study, we analyze how incorporating fire-induced heat and moisture release, together with aerosol–radiation interactions, affects atmospheric conditions in ICON-ART and how these changes ultimately influence the calculated plume height, without any tuning of the Freitas model itself. "

l264: "The calculation of aerosol radiative effects takes place in in the ICON-ART model. Aerosol-radiative effects impact the atmospheric stability close to the fire source, which is used by the plume rise model to calculate the emission height, as well as downstream of the plume. Other than the modified atmospheric profiles, the Freitas model does not account for aerosol-radiation interaction."

Further we added Section 3.1, which shows how ICON-ART variables are impacted by the implementations.

We believe these changes address your concerns and make the methodology more transparent and reproducible.

**The introduction felt a bit "long-winded," and I wonder if there's text that could be omitted. The authors may want to take a closer look at the content between lines 75–115 to see whether some material could be trimmed or written in a more concise and consolidated way.**

R: We trimmed and rewritten this section to:

l79:"Modeling aerosol radiative effects requires accurate representation of particle optical properties, which govern absorption, scattering, and their feedback on plume dynamics. Key parameters include single scattering albedo (SSA), extinction coefficient, and asymmetry factor, all of which vary with particle size, morphology, chemical composition, mixing state, and ambient humidity (Chen et al., 2006; Pang et al., 2023; Bond and Bergstrom, 2006; Bond et al., 2006; Petzold et al., 2005). These dependencies influence how aerosols interact with solar radiation, affecting heating rates and buoyancy, and thus plume evolution through processes like aerosol-induced lofting (Muser et al., 2020). Simplified assumptions, such as treating aerosols as spherical, externally mixed, and using fixed optical properties, introduce substantial uncertainties in radiative forcing estimates and feedbacks on smoke transport. For example, biomass burning aerosols in many climate models are too absorbing, leading to an overestimation of their warming effect (Brown et al., 2021). Improving model fidelity requires dynamic treatment of particle properties and coupling with meteorological conditions to capture vertical transport and long-range impacts accurately.

The lofting of the plume during the Australian New Year's event (ANY) illustrates the importance of these processes. Ma et al. (2024) showed that smoke rose in three distinct phases: initial pyro-convection up to 16 km, radiative self-lofting to 25 km driven by aerosol absorption, and a final ascent to 35 km facilitated by a stratospheric circulation vortex. These findings highlight how radiative effects can dominate vertical transport well beyond the initial injection height, reinforcing the need for models to incorporate such feedbacks."

**For the methodology section, it was unclear how moisture fluxes and aerosol radiative effects were implemented within the Freitas et al. plume-rise model. For example, was an aerosol-radiative-effect subroutine or parameterization added directly into the Freitas model? Or is smoke first lofted by the Freitas model and then inserted into the 3D fields of the ICON model, where aerosol radiative effects within ICON further loft the smoke aerosols through "radiative lofting"?**

R: Apologies for the confusion. There were no explicit modifications made to the Freitas plume-rise model—neither for moisture fluxes nor for aerosol–radiative interactions. The plume-rise model does not include any parameterization for aerosol radiative effects, and therefore there is no lofting due to aerosol radiative effects within the Freitas model. On the contrary, our results show that aerosol radiative effects near the surface tend to stabilize the atmospheric state. This stabilization influences the meteorological conditions provided to the Freitas model and, as a result, leads to lower calculated emission heights. All aerosol–radiative interactions and moisture flux implementations occur within the ICON-ART model. These processes modify the atmospheric fields (temperature, humidity, ...) that are read by the Freitas plume-rise model, indirectly affecting plume height calculations. The actual lofting due to radiative effects happens only in ICON-ART, not in the Freitas model.

l264: "The calculation of aerosol radiative effects takes place in in the ICON-ART model. Aerosol-radiative effects impact the atmospheric stability close to the fire source, which is used by the plume rise model to calculate the emission height, as well as downstream of the plume. Other than the modified atmospheric profiles, the Freitas model does not account for aerosol-radiation interaction."

**Along these lines, how were moisture fluxes incorporated into the Freitas et al. model? Were they used to modify the moisture content in the first vertical level of the Freitas model? This seems to be implied by the statement that "the moisture is added to the specific humidity tracer," but it's unclear whether this refers to ICON or Freitas. More details on how moisture fluxes were coupled with the Freitas model would be helpful.**

R: The moisture release is incorporated into the Freitas plume-rise model indirectly, through the atmospheric state provided by ICON-ART. In ICON-ART, moisture release from the fire is accounted for, and these modified meteorological fields are then read by the Freitas model. The vertical levels affected depend on the calculated emission height, as we assume the same vertical emission profile for moisture as for aerosols within ICON-ART. We believe the changes described in the first response and the following changed clarify this coupling.

l230: "The moisture emission follows the same injection height and profile as the particles, according to the plume-rise model. Therefore, the moisture-release emission rate $E_{qvfire}$ can obtained by inserting $qv_{fire}$ as $M$ in Equation 5. As, this additional moisture is added to the existing qv (specific humidity) tracer of ICON and thus, contributes to all physical and dynamical processes. The changes in the atmospheric state by either the heat or moisture release thereby change the atmospheric profiles that are inputs for the plume rise model, and therefore impact the calculated injection height."

**I also wonder whether Figure 1 could be used more effectively to describe the workflow and the order in which aerosol radiative effects and moisture fluxes are computed in the ICON/Freitas plume-rise framework. Figure 1 may also need to be reworked—it's a bit vague in its current form.**

R: We have reworked Figure 1 to make the workflow clearer and to explicitly described the order in which the implemented processes are computed. The figure now includes the relevant processes and their interactions, and the accompanying description has been adapted to clarify the workflow.
"Figure 1: Schematic illustration of the coupling between the plume-rise model and ICON-ART. The diagram shows the sequence of processes: fire-induced sensible heat release, emitted moisture, and aerosol–radiation interaction within ICON-ART, which modify the atmospheric state. These updated conditions are then passed to the Freitas plume-rise model for plume height calculation. Finally, aerosol and moisture emissions are applied, and the cycle repeats."

**The abstract and introduction discuss how fire–atmosphere interactions were implemented within the ICON/Freitas plume-rise modeling framework. I believe the manuscript uses the term "fire-induced meteorological effects." However, I did not see any explanation of how fire–atmosphere interactions were included in this framework. When I think about fire–atmosphere interactions, I think of models like WRF-FIRE or WRF-SFIRE, where the fire generates a plume, and through mass continuity the near-surface winds accelerate, causing the fire to spread faster and release more energy, which then further strengthens the plume-rise updraft. I suspect many fire modelers interpret "fire–atmosphere interactions" in this way. If these types of feedbacks are not included in the ICON/Freitas framework, the authors may want to revise the abstract to more clearly define what is meant by fire–atmosphere or meteorological interactions.**

R: We changed fire–atmosphere interactions to fire-induced meteorological changes.

**The structure of the discussion section could also be improved. Some of the descriptions seem to bounce around a bit, and it may be worthwhile for the authors to give this section another pass to make it read more coherently.**

R: We restructured the discussion, please see the changes in the revised manuscript.

**Specific comments:**

**Line 20: While I suspect this is not too far off, it's worth noting that these analyses were based on MISR data, which corresponds to satellite overpasses that occur during the late morning/early afternoon, when plumes are not fully developed. Granted, the PBL was also probably shallower during these times so the % above the PBL may not change a ton, but maybe just note that these overpasses occur in the late morning/early afternoon.**

R: We added the following statement:
"While most emissions remain within the planetary boundary layer (PBL), a significant fraction (4–20% over North America) reaches the free troposphere, enabling long-range transport and interaction with cloud systems (Kahn et al., 2007; Val Martin et al., 2010). These estimates are based on MISR observations during late-morning to early-afternoon overpasses, when plumes are typically less developed and the PBL shallower, a timing that should be considered in interpretation."

**Line 40: Another issue, especially for the empirical-based Sofiev scheme is that it's trained**

off MISR data and does a poor job with extreme plume rise values, e.g., the tail of the distribution. It does better with smaller plumes, but it's usually not the small plumes we care about. The larger fires that emit orders of magnitude more smoke are generally the ones that generate the plumes that loft smoke further into the free troposphere.

R: We added:
"A further limitation, particularly for the empirical Sofiev scheme, is its reliance on MISR-based training, which results in poor performance for extreme plume-rise events, which dominate smoke injection into the free troposphere."

**Line 60: I would argue the Freitas model is computationally efficient. As a standalone model, it runs in less than 0.1 seconds on a single CPU core. ML/AI approaches could be faster, but in terms of computational time relative to the host 3D model my guess is this would not really have much of an impact on the model run time.**

R: Thank you for pointing this out. We agree that the standalone Freitas model is computationally efficient. However, in our implementation within ICON-ART, the computational cost becomes significant because the plume-rise model is called for approximately 1,300 grid cells every time step. Assuming the reported runtime of 0.1 seconds per call on a single CPU core, this would amount to roughly 50 CPU node hours for one experiment. For comparison, a full experiment consumes about 120 CPU node hours, so the impact is notable in our setup. That said, we agree that this statement should be considered in context. To avoid confusion, we have removed the sentence from the manuscript.

**Line 160: replace "this limitation" with "cloud blocking", or something along these lines to make this clearer of what "this" is referring to.**

R: Done. "To address the data gaps caused by cloud blocking, fire data is assimilated using a Kalman filter (Rodgers, 2000)."

**Line 175: Slightly confused here, it was noted that geostationary satellites can observe the diurnal cycle, but it looks like an alternative cyclic function was used? It was unclear if this cyclic function was related to geostationary satellites observations or not.**

R: The function we used is an alternative, generally applicable approach and is not directly derived from geostationary satellite observations. We have clarified this in the text and included a more thorough discussion.

l169: "The diurnal cycle of fires varies considerably across regions and depends on both fuel type and meteorological conditions. For instance, extreme fires often show pronounced peak activity, extended duration throughout the day, and persistent nighttime burning. Despite these variations, a majority of vegetation fires follow a characteristic diurnal cycle that can be approximated by a Gaussian distribution, typically peaking in the early afternoon (Vermote et al., 2009). A generalized diurnal cycle function has been proposed by Kaiser et al. (2009) and Andela et al. (2015) and was applied by Walter et al. (2016) in COSMO-ART; this approach is also implemented in ICON-ART. The diurnal cycle function, denoted as $d(t_1)$, is applied to both fire intensity and fire size within the plume-rise model."

**I am sure this OK, but why not use a mass flux approach to directly compute the smoke detrainment (see Wilmot et al. 2022)? This gets away from the parabolic emission profile assumption, which is a little "hand wavy", and uses an approach based on plume physics. Wilmot, K., D. V. Mallia, A. G. Haller, and J. C. Lin, 2022: Wildfire plumes in the Western US are reaching greater heights and injecting more aerosols aloft under a changing climate. Scientific Reports, 12, 12400.**

R: We agree that the parabolic emission profile introduces uncertainty. We have included this point in the discussion section and now explicitly acknowledge that the assumed parabolic profile may not fully represent the actual vertical distribution of aerosols. However, when considering the fire's impact on meteorology, additional vertical transport within the host model enhances upward motion and aerosol transport toward the plume top, which may partially compensate for this discrepancy. We have added

a thorough discussion of these uncertainties .

l617: "The emission estimates are subject to several uncertainties. First, the emission profile, which is assumed to be parabolic, whereas studies such as (Moisseeva and Stull, 2021; Wilmot et al., 2022) indicate that the majority of aerosols are concentrated near the plume top. However, when considering the fire's impact on meteorology, additional vertical transport within the host model enhances upward motion and therefore aerosol transport toward the plume top, which may partially correct this discrepancy."

**Line 209: Where did the 3.4 GFAS correction factor come from? It looks like this is based on Kaiser et al., so are we just assuming that since the smoke emissions are off by a factor of 3.4, this difference is related to a 3.4 difference in fuel consumption, which would be proportional to heat fluxes? Is it possible that this 3.4 might be the result of uncertainty with fire emission factors? These emission factors, especially for PM2.5 can be quite large (see Urbanski 2014). It might be good to include some of the limitations of this assumption at the very least. Urbanski, Shawn. 2014. Wildland fire emissions, carbon, and climate: Emission factors. Forest Ecology and Management. 317: 51-60.**

R: This correction factor was empirically derived for particulate matter and may reflect uncertainties in the emission parameterization—which can be substantial, as noted by Urbanski (2014)—or a general underestimation of FRP values on which the parameterization is based. To further validate this approach, we compared direct MODIS FRP measurements with the FRP assumed in ICON, both with and without the 3.4 factor. This comparison demonstrates that the enhancement factor improves agreement with direct FRP observations. We have added a discussion of these limitations and cited Urbanski (2014) accordingly.

l635: "For particulate matter, Kaiser et al. (2012) shows that there is a systematical underestimation by GFAS and therefore introduces a correction factor of 3.4. We also applied this factor to our aerosol emissions, and based on Figure 10, the application of this factor shows overall good agreement with observations. This factor was empirically derived for particulate matter and may be attributed either to uncertainties in the emission parameterization, which can be substantial (Urbanski, 2014), or to a general underestimation of the FRP on which the parameterization is based. The comparison of direct MODIS FRP measurements with the FRP assumed in ICON, both with and without the 3.4 emission factor (see Appendix 2), demonstrates that applying the enhancement factor improves the agreement between the assumed FRP and MODIS observations. Although this does not constitute proof that the enhancement factor is universally applicable due to systematic underestimation, it can reasonably be assumed for the case considered here."

**Line 209: In equation 6, what is the 5.5? Did you mean 0.55?**

R: Please see l211: "The total energy released by fires is calculated by multiplying the Fire Radiative Power (FRP) by a factor of 10, as proposed by Val Martin et al. (2012) and applied in Ke et al. (2021). To estimate the portion of this energy contributing to convective processes, a factor of 0.55 is used, following Freitas et al. (2006)."

**Line 310: See the general comment, but it is unclear how moisture fluxes and aerosol–radiation interactions were incorporated in ICON/Freitas, along with the order of these calculations. It is also unclear what is meant by a fire-induced heat flux? If I remember correctly, the Freitas et al. plume rise model will not run if the Heat flux is equal to 0. If the "fire induced heat flux" is 0, is the plume rise model just being forced by terrestrial radiative fluxes?**

R: The fire-induced heat flux is implemented in the ICON-ART model, and modifies the atmospheric state there. The Freitas plume-rise model, however, runs independently of this heat flux. It is called based on the GFAS aerosol emission flux: if a grid cell exceeds a predefined emission threshold, it is considered a fire, and the plume-rise model is triggered. This same threshold is also applied for the new implementations of heat and moisture release in ICON-ART. Within the plume-rise model, the heat flux used for plume height calculation is determined by the vegetation type, as described in Section 2.1.1.

l133: "The Freitas plume-rise model is triggered in grid cells where the Global Fire Assimilation System (GFAS) aerosol emission flux exceeds $5\times10^{-12}$ kg m$^{-2}$. The plume-rise model reads in the meteorological conditions from ICON-ART, determines the heat flux according to the vegetation type within the grid cell and subsequently, returns the plume-top and bottom height to ICON-ART."

**Line 345: Are heights reported in ASL or AGL? Might be good to add this when reporting heights here, i.e., 3.5 kmAGL.**

R: We've added a statement in the "Definitions and analysis methods" section:

l360: "All heights refer to heights above sea level."

**Line 343: How does the fire destabilize the atmosphere in ICON? I do know that the Freitas model runs for X time steps to develop to build up the plume (I think this takes anywhere from 30-55 model minutes)? Is ICON reading output from each time step from the Freitas model and using that to modify the 3D weather/aerosol fields within ICON? Or are the emissions only released vertically once the plume has reached steady state? Along these lines, is the Freitas model one-way or two-way coupled with ICON, both in terms of meteorology/aerosols?**

Thank you for pointing this out. The term "fire intensity" refers to the diurnal cycle assumed within the plume-rise model, which peaks in the early afternoon. The phrase "decreasing stability" refers to the atmospheric profiles from ICON-ART that are read by the Freitas model and capture the gradual destabilization of the atmosphere throughout the day. We clarified this by changing the sentence to:
l498: "Since the ICON-ART meteorological input remains unaffected by the fire, the evolution of the calculated injection heights during the first day is driven by the typical diurnal cycle of atmospheric stability, which decreases throughout the day due to increased solar radiation. This destabilization is reflected in the ICON-ART atmospheric profiles used by the plume-rise model to calculate emission height, together with the diurnal cycle of fire intensity assumed in the plume-rise model. The combination of increasing fire intensity and decreasing atmospheric stability during the day leads to higher plume heights."

Additionally, the Freitas plume-rise model is called every simulation time step (one minute in our setup). It then calculates plume top and bottom either when steady state is reached or after a maximum duration of 200 minutes. During this time, the atmospheric conditions within Freitas are not updated. The coupling is one-way: ICON-ART provides meteorological input to Freitas, and Freitas returns only the emission height to ICON-ART. ICON does not read intermediate outputs from Freitas and does not modify its 3D weather or aerosol fields based on Freitas calculations. Emissions are released vertically once the plume height is determined.

**Line 424: The plume is rising– is this after the smoke has been vertically lofted by the Freitas model, i.e., has the smoke been carried upward by the Freitas model, vertically distributed within the ICON vertical column, and then lofted up further due the aerosol radiative heating that is resolved within ICON?**

R: Yes, that interpretation is correct. Figure 6 illustrates the semi-direct aerosol effect downstream of the plume within ICON-ART. The Freitas model only provides the initial injection height; it does not simulate any subsequent vertical lofting or cloud formation. These processes—such as aerosol radiative heating and cloud development—are resolved entirely within ICON-ART after the emissions have been injected at the height returned by the Freitas model. We have clarified this point in the revised text. Please see previous responses.

**Line 429-434: Near-surface aerosol layers promote stability by warming low/mid-level layers, which makes sense. But I am not entirely following the upper-level description? If the smoke is lofted further up, would there still be warming aloft and therefore a stable layer, which is just higher up in the atmosphere?**

R: Yes, your interpretation is correct. To clarify this point, we have added a new figure illustrating the vertical temperature changes associated with the aerosol layers and added following explanation:

l383: "Finally, we evaluate the aerosol–radiative effect. Figure 7 illustrates how aerosol–radiation interactions influence vertical wind velocity and temperature within the plume approximately one day after the simulation start. Figure 7 a shows that, in the ARI experiment, the probability of stronger updrafts and downdrafts increases, with a more pronounced enhancement in updrafts. This suggests that, compared to the REF experiment, the plume exhibits greater dynamical activity, as well as an overall lofting effect within the plume. Figure 7 b presents the mean temperature difference within the plume region. Below 3 km, an overall cooling effect is observed, whereas between 4 and 12 km, there is a warming effect, followed by cooling above 13 km. This pattern indicates warming within the plume core and cooling beneath it. The horizontal distribution of plume-top heights will be discussed later, however, it is important to note that, due to plume lofting, the top heights interacting strongly with radiation vary substantially: from about 2 km up to 16 km. Warming near the plume top is associated with cooling below, and thus spatial averaging significantly smooths these effects."

**Line 455: Okay, I think this loosely answers my earlier question about the order of operations for the aerosol radiative feedbacks. However, this really needs to be explained more clearly in the methods section. I strongly recommend redesigning Figure 1 so it clarifies the sequence of steps: the heat-flux/emission calculation - the Freitas model simulation - injection of smoke into ICON - the physics that ICON resolves once the smoke fields have been added. It seems that the same workflow applies to moisture as well.**

R: Please see the explanations above.

**While this order makes sense for aerosol radiative feedbacks, I wonder whether releasing moisture after the plume-rise calculation is a potential limitation? Moisture fluxes could be relevant for the plume-rise calculation itself. For example, if the lower boundary conditions in the Freitas model contained more moisture, the rising plume would reach the lifting-condensation level sooner, allowing latent-heat release to occur at a lower altitude. That additional buoyancy could enable the plume to rise higher. Coupled fire–atmosphere simulations have explored this effect, and although some preliminary results suggest that moisture fluxes may not matter much for plume development, I still wonder if adding moisture only after the plume-rise step is an oversimplification. This is, of course, assuming I'm interpreting the order of operations in the Freitas/ICON framework correctly.**

R: Yes, your interpretation of the order of operations is correct. Moisture is added after the plume-rise calculation. We acknowledge that this could be seen as an oversimplification. However, the plume-rise model is called every simulation time step (one minute in our setup), so moisture emissions from the previous time step are already included in the calculation. This means the spin-up phase for moisture effects is delayed by only one minute compared to sensible heat release, which we consider acceptable. The discrepancy between fire effects and plume calculation is therefore minimal. We have clarified the order of operations. See responses above.

**Line 470: Does ICON resolve SOA formation? Might be good to note this somewhere (maybe the methods, unless I missed this). Seems like there is no SOA based on the statement in line 563.**

R: No, a statement has been added to the model configurations.

l317: " However, no secondary organic aerosols accounted for."

**Line 535: This seems to be a common issue with plume-rise models and has been documented for the Freitas model as well. However, I'm not sure we can attribute the underestimation solely to the plume-rise model itself. The inputs to these models, namely heat flux and active fire area, are highly uncertain, and the model is very sensitive to heat-flux density. I'm not confident that we have a solid understanding of the actual heat-flux density for wildfires outside of a few field campaigns, since current satellite fire detection data is likely too coarse to definitively determine whether plume-rise models truly have a systematic underestimation problem. Maybe we are just systematically underestimating the heat flux density? This point goes beyond the scope of the paper, and satellite-derived**

**inputs remain one of the best available option despite their limitations. Still, it may be worth noting this somewhere in the discussion or conclusion section. Along these lines, while the addition of moisture and aerosol physics within the Freitas/ICON framework appears to improve the results, a major caveat is that these enhancements may be compensating for systematic underestimation of how plume-rise inputs (heat flux and fire area) are computed. In any case, this isn't something I expect the authors to resolve—it's just a thought. I believe this idea is somewhat alluded to around line 546.**

R: Thank you for this remark. We included this in the discussion:

l609: "While underestimation of plume heights has been documented for plume-rise models, including Freitas, attributing this bias solely to the parameterization may be overly simplistic. Heat-flux density—a key driver of plume buoyancy—is poorly constrained outside of limited field campaigns. Satellite-derived fire products, which provide the majority of input data, may systematically underestimate this quantity, complicating the interpretation of model–observation differences. Enhancements such as fire-induced meteorological changes within the ICON framework may partly compensate for low-biased inputs rather than exclusively correcting physical parameterizations. However, this study aims to develop a model setup that can be applied globally and in near real time, and is therefore restricted to suitable available datasets and the information they provide."